# Deep Evolutionary Learning for Molecular Design

## Abstract

In this paper, we propose a deep evolutionary learning (DEL) process that integrates fragment-based deep generative model and multi-objective evolutionary computation for molecular design. Our approach enables (1) evolutionary operations in the latent space of the generative model, rather than the structural space, to generate novel promising molecular structures for the next evolutionary generation, and (2) generative model fine-tuning using newly generated high-quality samples. Thus, DEL implements a data-model co-evolution concept which improves both sample population and generative model learning. Experiments on two public datasets indicate that sample population obtained by DEL exhibits improved property distributions, and dominates samples generated by multi-objective Bayesian optimization algorithms.

## 1 Introduction

A drug is a molecule that binds to a target (e.g. protein) to inhibit or activate specific pathways in pathogens or host cells that cause abnormal phenotype. Drug discovery and development is a costly and time-consuming process, which is compounded by personalized medicines development for cancer or other complex and rare diseases. Computational drug discovery has been shown to accelerate the whole discovery process using simulations and machine intelligence. However, challenge remains in this field by demands for a robust and unbiased feature representation theories for molecules and their corresponding receptors, and efficient search algorithms. The rise of AI and data science provides us with a unique opportunity to reevaluate the problem and develop fast intelligent search or design approaches (Gromski et al., 2019; Chen et al., 2018). These new technologies claim differences from the traditional ones in two aspects: (1) features can be automatically learned using embedding techniques on a large number of training samples, and (2) high-level relationships (in supervised case) and complex distributions (in unsupervised case) can be captured using appropriate deep architectures. Recently, new representation theories and architectures have been proposed in the domain of molecular generation. There exist two new major methods to present a molecule for a machine learning algorithm. The first method converts a molecule structure to a string, such as the simplified molecular-input line-entry system (SMILES) string (Weininger, 1988), and adopt natural language processing (NLP) methods for supervised or unsupervised learning. The second uses an undirected graph to present a molecular structure and applies graph convolutional neural networks (Duvenaud et al., 2015). Three major families of AI algorithms have been developed for novel drug discovery: deep generative models (DGMs), reinforcement learning, and the combination of both.

As one family of major neural probabilistic models for data modelling, generative autoencoders (e.g. variational autoencoder (VAE) (Kingma & Welling, 2014)) have been adopted to learn on either SMILES strings (Romez-Bombarelli et al., 2018) or molecular graphs (Simonovsky & Komodakis, 2018) with corresponding physical and biochemical properties for molecular generation. By integrating the generative adversarial nets and autoencoders, adversarial autoencoders (AAEs) have also been well applied to molecular design (Kadurin et al., 2017). The advantage of using generative autoencoders is that, molecules, as discrete objects in our world, are mapped to the continuous latent space, whose landscape can be organized by their properties, which helps generate new structures with preferred property values. However, critical problems remain due to imperfect representation methods. When SMILES strings are used in VAE, the model suffers from imbalance of tokens in embedding, generation of invalid structures, and the problem where two almost identical molecules have markedly different canonical SMILES strings. When using graph as molecular

representation in VAE, a technical difficulty is to design an effective graph decoder. In addition to other heuristic methods, a SMILES decoder can be used to pair with a graph encoder. While DGMs offer convenience of searching in latent space, reinforcement learning algorithms can directly search in the molecules' structural space by adding or deleting bounds and atoms (Zhou et al., 2019). In the Markov decision process (MDP) for drug design, the agent is a molecular generator, the molecular structure indicates the state, the actions are modifications to the current structure, and a simulator (e.g. surrogate model) is often used as the environment to provide reward. Furthermore, generative and predictive models can be integrated in MDP to form deep reinforcement learning (DRL) methods, where the generative model is trained as a policy approximation and the predictive model can be used as a value function approximation (You et al., 2018; Popova et al., 2018). Search in the discrete input space and inefficient learning are arguable concerns to be addressed when applying reinforcement-learning-based solutions for compound design.

Interestingly, as an old peer of reinforcement learning for black-box optimizations, evolutionary computation (EC) methods (Eiben & Smith, 2015) have been catching up with promising performances in modern optimization, design and modelling problems. Besnard et al. (2012) present a strategy for evolution of ligands along multiple properties in the structural space, where a library of knowledge-based chemical structural transformation is used as the mutation operator. Interactions between EC and neural networks have mainly focused on network evolution and neural surrogate models for fitness functions. For examples, EC has been used at a large scale for neuroevolution that leads to evolution of neural network architectures (Stanley et al., 2019); feedforward neural network is commonly used as fitness function in EC (Mandal et al., 2019). Furthermore, it has been recently discovered that evolutionary strategy (ES) can perform competitively with reinforcement learning in game AI (Salimans et al., 2017). ES (Wierstra et al., 2014) and estimation of distribution algorithms (EDA) (Hauschild & Pelikan, 2011) build parameterized search distributions over promising points and either employ gradient information or sample from such a probabilistic model to find better points. Both probabilistic strategies from EC can potentially be used as alternatives to Bayesian optimization (BO) in (continuous) black-box optimizations. A single-objective BO has been recently applied to molecule optimization in the latent space of VAE (Romez-Bombarelli et al., 2018).

Since model learning is essentially parameter estimation from the statistical modelling perspective, the quality of data in deep learning is crucial for model performance. Data augmentation is becoming a new strategy in deep learning to improve the training of a model. For example, in computer vision, transformations (such as rotation and flipping) of images are used to increase the sample size when the original data set is insufficient (Perez & Wang, 2017; Cubuk et al., 2019; Shorten & Khoshgoftaar, 2019). In NLP, a text dataset can be augmented using tricks such as replacing words or phrases with their synonyms (Wei & Zou, 2019), and resorting aids from other language models (e.g. word embedding and neural machine translations) (Sennrich et al., 2016; Wei & Zou, 2019). Basically, these methods either increase the data by transforming existing information which can only alleviate the limit of certain techniques (e.g. convolution), or indirectly borrow new information from other sources (e.g. methods in NLP).

In summary, even though modern machine learning, evolutionary computation, and data science methods have been applied to molecular design and achieved promising results, we are still challenged by three chief issues. (1) An effective representation method for compound structures, that is encoding-decoding friendly and invariant to multimorphic forms, is still missing. (2) New ideas are expected for effective representation and coding of discrete structures in EC. And, (3) quality of data can be further improved as current data augmentation tricks only increase the number of samples but does not specifically address data quality. In this paper, we propose a novel deep evolutionary learning (DEL) process that combines the merits of deep generative model and multi-objective evolutionary computation together for molecular design. Specifically, our work has three major contributions. (1) In our approach, latent representations of phenotypic samples in a population serve as genotypic codes for evolutionary operations.This approach differs from traditional evolutionary algorithms that search in the original space of a problem. Specifically, our framework's DGM encoder projects molecular structures in a population from discrete space to continuous latent space where evolutionary operations are applied to help explore the latent representation space. Subsequently, the DGM decoder maps the genotypic representation to the phenotypic space for generating new molecules with desired property values. (2) In each evolutionary generation, the newly formed population containing novel competitive molecules can be used to further fine-tune the DGM. This approach is an innovative data augmentation strategy that enriches training data with novel high-

quality samples. The whole DEL process implements a new learning paradigm that co-evolves data and model alternatingly through multiple evolutionary generations. (3) Our comprehensive experiments demonstrate that DEL is able to produce populations of novel samples with improved values of properties and outperforms state-of-the-art multi-objective BO algorithms (MOBO).

## 2 METHOD

The proposed deep evolutionary learning (DEL) process combines deep learning and multi-objective EC through the latent representation space of molecules. One of the theoretical innovations of our approach is that it demonstrates that EC methods are extendable to corresponding deep versions. The main idea is illustrated in Figure 1a and formally presented in Algorithm 1 (see Appendix A.1). This algorithm consists of the following steps. (a) A VAE (as molecule modeller) and a multi-layer perceptron neural network (MLP, property predictor as regularizer) are pretrained using all the original training data to start the first evolutionary generation, or, if not in the first generation, using samples from the previous population. (b) Training samples (if first generation) or population samples (otherwise) are projected to the latent space using the encoder of the VAE. (c) Based on non-dominated ranking and crowding distances of samples with respect to multiple properties, evolutionary operations (selection, recombination/crossover and mutation) are conducted on the latent representations of the samples. (d) Given these new latent codes after evolutionary operations, new molecule samples are generated by the decoder of the VAE. (e) Properties of these generated samples are obtained using a simulator (e.g. RDKit (Landrum, 2006) in our experiment). (f) New samples with good desired properties and good samples from the previous generation form the new population. (g) Steps (b-f) are iterated for multiple generations. (h) The final population is returned.

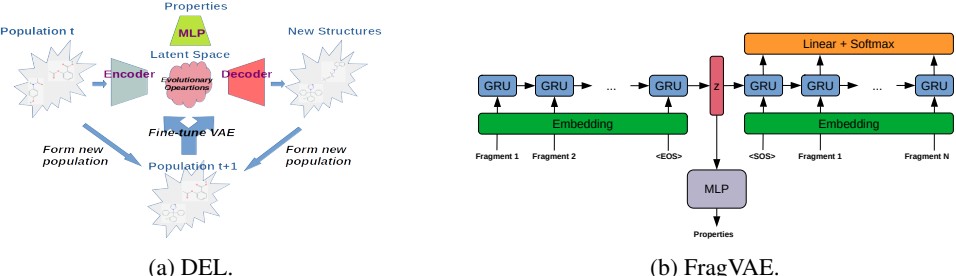

(a) DEL.             (b) FragVAE.

Figure 1: Deep evolutionary learning process and deep generative model integrated in DEL.

The major advantages of our DEL algorithm over existing interactions between EC and neural networks can be explained as follows. (a) We directly evolve a collection of data rather than many neural network structures and parameters. Data evolution tends to be more efficient than direct evolution of model structures and parameters. (b) The single neural network model (i.e. DGM in DEL) can be indirectly improved through learning on the evolved data with modern gradient-based variational learning and inference algorithms. Thus, the improvement of populations along evolutionary generations can be viewed as an effective data augmentation strategy that includes novel and high-quality samples for further training of the neural network. (c) The continuous latent representation space established by the encoder of DGM can be naturally used as encoding (genotypic) space for evolutionary computation. Thus, evolutionary operations are carried out in the latent space instead of the discrete structural input space, allowing more efficient and smooth exploration, because the latent space is often multimodal and can be organized by properties (regularized by the property predictor) and evolutionary operations in this space can help the search escape from local regions and explore new regions of interest. (d) The multi-objective operations - non-dominated sorting and crowding distance, can help identify competitive and diverse parent samples to breed offspring. In summary, DEL takes advantages of both multi-objective EC and probabilistic neural model learning. The DGM, multi-objective components (non-dominated sorting and crowding distance), evolutionary operations, formation of new populations are discussed in details as below.

### 2.1 FRAGVAE FOR FRAGMENT-BASED MOLECULAR MODELLING

In our DEL process, we adopted a VAE model originally for fragment-based molecular generation (Podda et al., 2020). The concept of fragment-based drug design (FBDD) was introduced in

(Shuker et al., 1996). In FBDD-based approaches, small organic molecules that bind to proximal subsites of a protein are identified, optimized, and linked together to produce high-affinity ligands. Wet-lab approaches for FBDD include X-ray crystallography and NMR spectroscopy. Compared to atom-based drug design, FBDD has the following advantages (Erlanson, 2011). (1) The search space in FBDD is much smaller ($10^7$ versus $10^{60}$). (2) Identifying a fragment with certain affinity to the target may mean finding a pharmacophore. (3) Fragment-based synthesis could be more efficient than high-throughput screening. Majority fragmentation methods, which break a molecule into parts, are based on synthetic accessibility. For examples, RECAP (retrosynthetic combinatorial analysis procedure) is a method that breaks bonds formed by chemical reactions (Lewell et al., 1998); BRICS (breaking of retrosynthetically interesting chemical substructures) generates a more elaborated set of fragmentation rules along synthetically accessible bonds and generates more fragments than RECAP (Degen et al., 2008). BRICS is used in (Podda et al., 2020) to chop a SMILES string into several fragments. Then, fragment embeddings are produced using Word2Vec (Mikolov et al., 2013). Next, the sequences of fragments are modelled by a GRU-based VAE. In our work, a multi-head feedforward neural network component for predicting values of properties is added to the original model such that the latent representations can be regularized by properties of interest. Additionally, we normalize the three loss terms using batch size, and allow tuning of the weights among the loss terms. A crucial implementation bug in the original VAE model was also corrected (see Appendix A.2). Hereafter, we name this modified VAE model for fragments as FragVAE whose architecture is displayed in Figure 1b.

We denote the encoder parameter by $\phi$, the decoder parameter by $\theta$, and the property predictor network by $f_\psi(z)$ parameterized by $\psi$. The objective (to be minimized) of this DGM employed in DEL is a weighted combination of three terms:

$$l_{\phi,\theta,\psi} = -\mathrm{E}_{q_\phi(z|x)}[\log p_\theta(x|h)] + \beta \mathrm{KL}\big(q_\phi(z|x)||p_\theta(z)\big) + \alpha \mathrm{E}_{q_\phi(z|x)}\big[\mathrm{MSE}(f_\psi(z), y)\big], \quad (1)$$

where the first term is to reduce the reconstruction error, the second term is to regularize the posterior latent distribution with a simple prior, and the third term uses mean squared error of property prediction to further regularize the posterior distribution of latent codes. Previous studies unveil that VAE can easily fail on modelling text data because of the training imbalance between the reconstruction error (difficult to reduce once the KL divergence becomes very small) and the KL divergence (easy to diminish to zero). Thus, proper trade-off between the reconstruction error and KL divergence through $\beta$-VAE (Higgins et al., 2017) is vital in text generation and molecular generation (Yan et al., 2020; Bowman et al., 2016). In practice, the value of $\beta$ should be smaller than 1. To look for a suitable value of $\beta$, we design a versatile function, called $\beta$-function, as formulated below,

$$\beta(t) = \min\left\{\max\left\{ae^{k(1-\frac{T}{t})}, l\right\}, u\right\}, \quad (2)$$

where $T$ represents the total number of epochs, $t \in \{1, 2, \cdots, T\}$ indicates the current index of epoch, $k$ controls the incremental speed, $a$ defines the amplitude, $l$ and $u$ serve as lower and upper bounds respectively for the value of $\beta$. With different settings, a variety of curves of this function are shown in Figure 6 (see Appendix A.7). The value of $\alpha$ can be set similarly in FragVAE.

## 2.2 Non-domination Rank and Crowding Distance

To get non-domination rank and crowding distance of a feasible solution for guiding the sample selection (Section 2.3) and the population merging (Section 2.4), the fast non-dominated sort and crowding comparison methods are adopted from the classic NSGA-II algorithm for multi-objective optimization (Deb et al., 2002). The properties in molecular design are treated as objectives. In an optimization problem with $K$ objectives $f(z) = \{f_1(z), \cdots, f_K(z)\}$, feasible solution $z_1$ is said to dominate $z_2$ (denoted by $z_1 \prec z_2$), if $\forall k \in \{1, \cdots, K\}$: $f_k(z_1) \leq f_k(z_2)$ and $\exists k \in \{1, \cdots, K\}$: $f_k(z_1) < f_k(z_2)$. Using this concept of domination, all feasible solutions in a collection can be sorted to form Pareto frontiers (or fronts, ranks) $\mathcal{F} = \{\mathcal{F}_1, \mathcal{F}_2, \cdots\}$. Samples in the same frontier do not dominate each other. Frontier $\mathcal{F}_i$ dominates $\mathcal{F}_j$ for $j > i$. Thus, we define function $F(z)$ to retrieve the rank (i.e. frontier index) of any feasible solution $z$ in the population.

The crowding distance of a feasible solution is computed as the normalized perimeter of the cuboid formed by its immediate neighbours along all objective axes. To compute the crowding distance of $z_i$, the normalized distance between its nearest neighbours above (denoted by $z_a$) and below (denoted by $z_b$) it w.r.t. the $k$-th objective axis is calculated using $d_k(z_i) = \frac{f_k(z_a) - f_k(z_b)}{f_k^{\max} - f_k^{\min}}$, where

$f_k^{\max}$ and $f_k^{\min}$ are respectively the maximal and minimal values of the $k$-th objective. Then, these individual results are summed up to form the crowding distance of $z_i$: $d(z_i) = \sum_{k=1}^{K} d_k(z_i)$. The concept of crowding distance measures the density of the area around a feasible solution.

Using the two concepts, partial order can be defined. We say $z_1 \prec_n z_2$ if either (1) $z_1 \prec z_2$ (that is $F(z_1) < F(z_2)$), or (2) $F(z_1) = F(z_2)$ and $d(z_1) > d(z_2)$. When two solutions have same rank, the one with larger crowding distance is preferred because it helps maintain a diverse population.

### 2.3 EVOLUTIONARY OPERATIONS

The evolutionary operations include parent selection, recombination and mutation to produce new offspring in the evolutionary process. Binary tournament selection is applied to select one out of two randomly drawn samples from the current population. In such a selection process, let us suppose $z_1$ and $z_2$ are randomly taken from the population and $z_1 \prec_n z_2$. Sample $z_1$ will be selected with selection probability $p_s$ which is close to one, and $z_2$ will be selected with a small chance $1-p_s$. This selection process is repeated $M$ times to thus find $M$ parents where $M$ is the fixed population size. A pair of such parents will produce two children through recombination and mutation operations. Given two parents' latent representation $z_{p1}$ and $z_{p2}$, there are two recombination options - linear and discrete methods, to produce their new children $\hat{z}_1$ and $\hat{z}_2$. For linear recombination, $\hat{z}_1 = z_{p1} + r_1(z_{p2} - z_{p1})$ and $\hat{z}_2 = z_{p1} + r_2(z_{p2} - z_{p1})$ where $r_1 = -d + (1+2d)\alpha_1$, $r_2 = -d + (1+2d)\alpha_2$, $d = 0.25$, and $\alpha_1, \alpha_2 \sim$ Uniform(0,1). For discrete method, supposing a latent representation vector is of length $L$, an integer $l$ is randomly drawn from $\{1, \cdots, L-1\}$ such that $\hat{z}_1 = [z_{p1}[1 : l], z_{p2}[l+1 : L]]$ and $\hat{z}_2 = [z_{p2}[1 : l], z_{p1}[l+1 : L]]$. After crossover, a new sample $\hat{z}_m$ ($m \in \{1, \cdots, M\}$) will have a small mutation probability $p_m$ (say 0.01) of getting mutation. For $\hat{z}_m$, a random value $r$ is drawn from Uniform$(0, 1)$. If $r < p_m$, then a random integer $l$ is randomly selected from $\{1, \cdots, L\}$ such that the $l$-th position of $\hat{z}_m$ is replaced with a value drawn from standard Gaussian distribution: $\hat{z}_{m,l} \sim \mathcal{N}(0, 1)$.

### 2.4 FORMING NEW POPULATION

Ideally we need to maintain excellent and diverse populations. After possible mutation operations, all $M$ genotypic coding vectors will pass through the decoder of the DGM to produce phenotypic samples. All valid samples (supposed in set $\hat{\mathcal{P}}_{t+1}$) will be kept to merge with the previous population (denoted by $\mathcal{P}_t$) to produce a new generation (denoted by $\mathcal{P}_{t+1}$). To implement it, all samples in $\hat{\mathcal{P}}_{t+1} + \mathcal{P}_t$ are sorted based on their non-domination ranks first and then on their crowding distances. Finally, only the top $M$ samples are taken from them to form the new population.

## 3 EXPERIMENTS

The performance of DEL was investigated on the ZINC (Irwin & Shoichet, 2005) and PCBA (Wang et al., 2016) datasets. These data were processed in the work of Podda et al. (2020). ZINC and PCBA are respectively composed of 227,945 and 383,790 molecules with two or more fragments. More statistics of both data can be found in (Podda et al., 2020). We comprehensively investigated the empirical performance of FragVAE and DEL. Three properties (QED: quantitative estimation of drug-likeness, SAS: synthetic accessibility score, and logP: water-octanol partition coefficient) are selected as objectives in DEL. Molecules with large QED, low SAS, and small logP values are prioritized. Incorporation of other properties (e.g. binding affinity, structure-property relationship, and ADME) will be considered in future work. QED, a scalarization of eight molecular properties (including logP) (Bickerton et al., 2012), is an adequate initial screening step for drug candidates. As we dive into more specific applications, selective properties can be tailored for subsequent screening. Thus, explicit usage of logP as one objective in DEL can help better assess lipophilicity, a key factor in drug design for some diseases, e.g. kidney and heart problems.

### 3.1 EVALUATION OF FRAGVAE

As FragVAE is a significant modification of the original model used in (Podda et al., 2020), we investigated the impact of $\beta$ value to the performance of FragVAE in terms of loss function values

through Figure 7 (see Appendix A.7). Other hyperparameter values can be found in Appendix A.3. One can see that a large $\beta$ value can quickly reduce the KL loss to near zero which leads to stagnant reductions of reconstruction error and property regression error – the notorious posterior collapse problem (Goyal et al., 2017), because it is much easier to reduce the KL divergence than the reconstruction error in complex sequence modelling. Using a suitable small value of $\beta$ would allow the continuous decrease of the reconstruction error and property regression error. This observation is consistent with discoveries in language generative models (Yan et al., 2020; Bowman et al., 2016).

Table 2 (see Appendix A.6) shows the validity, novelty and diversity of 20,000 samples generated from trained FragVAEs using standard Normal prior to sample $\boldsymbol{z}$ followed by the decoder. Results of previous language-model-based and graph-based methods are also given for comparison. In general, the validity is defined as the ratio of number of valid generated samples to total number of generated samples. To clarify, the perfect validity reported in Podda et al. (2020) is actually calculated as the ratio of valid generated SMILES strings after discarding invalid fragment sequences versus total number of valid fragment sequences, i.e. *Validity (SMILES)* in Table 2. We found that this ratio is always 1 in fragment-based models. To have a better understanding about the model, we hence computed the validity of fragment sequences as the percentage of number of valid fragment sequences to total number of generated fragment sequences, i.e. *Validity (Fragments)* in Table 2. The novelty is defined as the ratio of number of generated novel valid molecules that do not exist in the training data versus total number of generated valid samples. The diversity is calculated as the percentage of generated unique valid samples among total number of generated valid samples. When the value of $\beta$ is very small (0.01), the posterior $p(\boldsymbol{z}|\boldsymbol{x})$ is highly different from the simple standard Normal prior $p(\boldsymbol{z})$. Thus, it is reasonable to see relatively low diversity in samples derived using standard Normal distribution. However, it does not imply that FragVAE with a very small value of $\beta$ is poor at learning latent representation. In fact, previous work in $\beta$-VAE shows that small values of $\beta$ tend to encourage disentangled representations and form latent clusters (Li et al., 2020).

The property distributions of generated samples can be important indicators to the proximity of generated samples with actual samples. Figure 2 shows the distributions of QED, SAS and logP in samples generated using standard Normal prior. Large $\beta$ value can lead to the sample SAS distribution appearing at the right side of the actual SAS distribution. Interestingly, the property distributions when using $\beta = 0.01$ do not resemble actual data, implying that using a very small value of $\beta$ would lead to latent representations that deviate from standard normal distributions. To summarize, $\beta \leq 0.1$ can prevent the model training from posterior collapse and can form structured latent representation space which is useful for latent space exploration using optimization techniques.

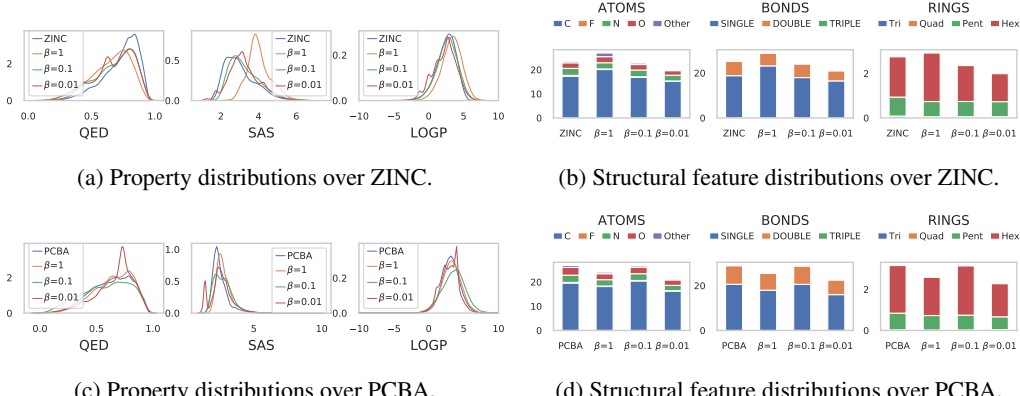

(a) Property distributions over ZINC.

(b) Structural feature distributions over ZINC.

(c) Property distributions over PCBA.

(d) Structural feature distributions over PCBA.

Figure 2: Property and structural feature distributions over 20K randomly sampled molecules.

## 3.2 DEL PERFORMANCE

We executed DEL processes using fixed or annealed loss trade-off weights: (1) $\beta = 0.1$ and $\alpha = 1$, (2) $\beta = 0.01$ and $\alpha = 1$, (3) $\beta = 0.1$ and $\alpha = 1$ in the initial training of FragVAE, then annealing $\beta$ to 0.4 and $\alpha$ to 4 in the second generation of DEL (denoted by $\beta = 0.1 \rightarrow 0.4$, $\alpha = 1 \rightarrow 4$), and similarly (4) $\beta = 0.01 \rightarrow 0.4$, $\alpha = 1 \rightarrow 4$. Other hyperparameter values can be found in Appendix A.4. The change of losses is shown in Figure 8. We observe that (1) all settings lead to similar reconstruction error convergence, (2) smaller values of $\beta$ tend to obtain smaller property

prediction error, (3) annealing can help reduce the KL divergence compared to the corresponding fixed settings.

Figure 3 shows population validity, novelty and diversity in different DEL processes. In this chart, *validity (SMILES)* is the validity of SMILES strings in the population; *validity (fragments)* is the validity of fragment sequences sampled using FragVAE after evolutionary operations; *novelty* is the ratio of population samples that are not in the training data against the population size; and *diversity* is the ratio of unique population samples against the population size. We observe that almost all samples in the populations are novel; and the population samples in the last generation are quite diverse, ranging from 0.798 to 0.988. Table 9 lists the increasing numbers of *high-quality* novel molecules discovered along the DEL processes. In contrast to the training molecules visualized in Figures 9 and 10 (Appendix), DEL is able to discover novel and diverse

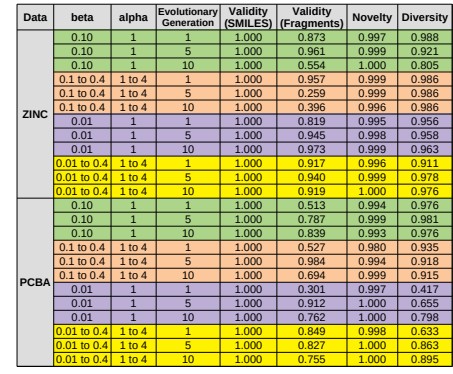

| Data | beta | alpha | Evolutionary Generation | Validity (SMILES) | Validity (Fragments) | Novelty | Diversity |
|------|------|-------|-------------------------|-------------------|----------------------|---------|-----------|
| ZINC | 0.10 | 1 | 1 | 1.000 | 0.873 | 0.997 | 0.988 |
| | 0.10 | 1 | 5 | 1.000 | 0.961 | 0.999 | 0.921 |
| | 0.10 | 1 | 10 | 1.000 | 0.554 | 1.000 | 0.805 |
| | 0.1 to 0.4 | 1 to 4 | 1 | 1.000 | 0.957 | 0.999 | 0.986 |
| | 0.1 to 0.4 | 1 to 4 | 5 | 1.000 | 0.259 | 0.999 | 0.986 |
| | 0.1 to 0.4 | 1 to 4 | 10 | 1.000 | 0.396 | 0.996 | 0.986 |
| | 0.01 | 1 | 1 | 1.000 | 0.819 | 0.995 | 0.956 |
| | 0.01 | 1 | 5 | 1.000 | 0.945 | 0.998 | 0.958 |
| | 0.01 | 1 | 10 | 1.000 | 0.973 | 0.999 | 0.963 |
| | 0.01 to 0.4 | 1 to 4 | 1 | 1.000 | 0.917 | 0.996 | 0.911 |
| | 0.01 to 0.4 | 1 to 4 | 5 | 1.000 | 0.940 | 0.999 | 0.978 |
| | 0.01 to 0.4 | 1 to 4 | 10 | 1.000 | 0.919 | 1.000 | 0.976 |
| PCBA | 0.10 | 1 | 1 | 1.000 | 0.513 | 0.994 | 0.976 |
| | 0.10 | 1 | 5 | 1.000 | 0.787 | 0.999 | 0.981 |
| | 0.10 | 1 | 10 | 1.000 | 0.839 | 0.993 | 0.976 |
| | 0.1 to 0.4 | 1 to 4 | 1 | 1.000 | 0.527 | 0.980 | 0.935 |
| | 0.1 to 0.4 | 1 to 4 | 5 | 1.000 | 0.984 | 0.994 | 0.918 |
| | 0.1 to 0.4 | 1 to 4 | 10 | 1.000 | 0.694 | 0.999 | 0.915 |
| | 0.01 | 1 | 1 | 1.000 | 0.301 | 0.997 | 0.417 |
| | 0.01 | 1 | 5 | 1.000 | 0.912 | 1.000 | 0.655 |
| | 0.01 | 1 | 10 | 1.000 | 0.762 | 1.000 | 0.798 |
| | 0.01 to 0.4 | 1 to 4 | 1 | 1.000 | 0.849 | 0.998 | 0.633 |
| | 0.01 to 0.4 | 1 to 4 | 5 | 1.000 | 0.827 | 1.000 | 0.863 |
| | 0.01 to 0.4 | 1 to 4 | 10 | 1.000 | 0.755 | 1.000 | 0.895 |

Figure 3: Population validity, novelty and diversity during DEL processes.

high-quality molecules (see Figures 11-18 in Appendix). Furthermore, the distributions of properties and structural features of samples along the evolutionary process are compared in Figure 4 (and Figures 19-21 in Appendix). It can be seen that the process can be quite different from the actual data distribution and is able to gradually improve the distributions of QED, SAS and logP in population samples towards the preferred goals. Interestingly, samples randomly generated using standard Normal prior do not clearly show this trend (Figures 22-25 in Appendix).

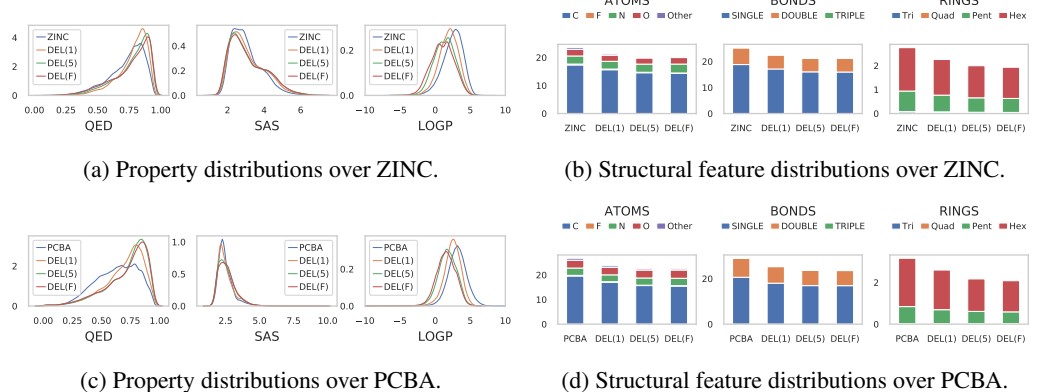

(a) Property distributions over ZINC.

(b) Structural feature distributions over ZINC.

(c) Property distributions over PCBA.

(d) Structural feature distributions over PCBA.

Figure 4: Property & structural feature distributions of DEL populations ($\beta=0.01 \rightarrow 0.4$, $\alpha=1 \rightarrow 4$).

### 3.3 COMPARISON WITH MULTI-OBJECTIVE BAYESIAN OPTIMIZATION

DEL was compared with two MOBO methods: q-Pareto Efficient Global Optimization (qParEGO) and q-Expected Hypervolume Improvement (qEHVI) (Daulton et al., 2020). These MOBO methods were run in the latent space of FragVAE trained in the first generation of DEL using all training samples. Hyperparameter settings of these algorithms are listed in Appendix A.5. Figure 26 shows the hypervolumes of both algorithms and the quasi-random baseline which selects candidates from a scrambled Sobol sequence along batches. Hypervolumes obtained using these models are plotted in Figure 26. It shows that qParEGO and qEHVI work better with $\beta = 0.01$ than that with $\beta = 0.1$.

To qualitatively compare DEL with the MOBO algorithms, the first five Pareto fronts obtained using DEL and last five batches obtained using qParEGO and qEHVI are visualized in Figure 5 and Figure 27 in Appendix. We can see that the solutions from qParEGO and qEHVI are behind the Pareto fronts from DEL. To quantitatively compare DEL with qParEGO and qEHVI, we conducted non-dominated sorting on the combination of the first Pareto front of DEL and the last six batches of qParEGO and qEHVI and report the results in Table 1. It shows that all solutions from the DEL Pareto front stay in the new integrated Pareto front, while almost all solutions from qParEGO and qEHVI are behind the integrated Pareto front. Furthermore, as indicated in Table 3 in Appendix,

DEL runs more efficiently than these MOBO algorithms, even though the population size (20,000) of DEL is much larger than the batch sizes (8) of MOBO algorithms. It has been a well-known challenge to scale up BO algorithms.

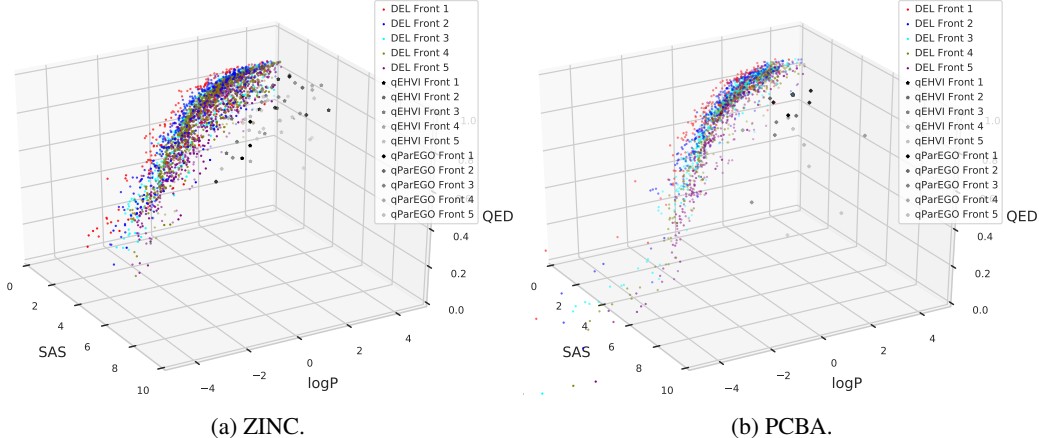

(a) ZINC.                                                           (b) PCBA.

Figure 5: Pareto fronts of DEL and MOBO algorithms ($\beta = 0.01 \rightarrow 0.4$). In the legend, the last batch of qParEGO or qEHVI is written as Front 1.

Table 1: Non-dominated sorting of combination of DEL, qParEGO and qEHVI Pareto fronts.

| | | Pareto Front Size for Comparison | | | In New Pareto Front | | |
|---|---|---|---|---|---|---|---|
| Data | DEL Hyperparameter | DEL | qParEGO | qEHVI | DEL | qParEGO | qEHVI |
| ZINC | $\beta = 0.1 \rightarrow 0.4$ | 243 | 46 | 45 | 243 (100%) | 0 (0%) | 0 (0%) |
| ZINC | $\beta = 0.01 \rightarrow 0.4$ | 200 | 46 | 40 | 200 (100%) | 0 (0%) | 1 (2.5%) |
| PCBA | $\beta = 0.1 \rightarrow 0.4$ | 228 | 46 | 36 | 228 (100%) | 0 (0%) | 0 (0%) |
| PCBA | $\beta = 0.01 \rightarrow 0.4$ | 183 | 46 | 43 | 183 (100%) | 1 (2.17%) | 2 (4.65%) |

## 3.4 ABLATION STUDIES

By default, DEL uses the property predictor for latent representation regularization, fine-tunes Frag-VAE using new population data in each generation, forms new child latent codes using the linear crossover method, and fixes the population size to 20K. Variants were created by (1) disabling the property predictor, (2) disabling finetuning, (3) using the discrete crossover method, and (4) allowing a much larger population size (100K). While it is difficult to compare these variants in terms of validity, novelty and diversity of population samples (see Figure 28 in Appendix), it turns out that non-dominated sorting is an informative method for comparison. Table 4, in Appendix A.6, indicates that DEL with the property predictor outperforms the variant without it. Table 5 shows that FragVAE finetuning in DEL can help obtain better Pareto fronts. Table 6 implies that both linear and discrete crossover operations behave well in DEL. Also, DEL with larger population size can form better Pareto fronts (see Table 7). Additionally, we applied non-dominated sorting to compare the quality of Pareto fronts obtained using different values of $\beta$ on DEL, and found that the integrated Pareto front consists of samples relatively evenly from all settings (Table 8).

## 4 CONCLUSION

In this paper, we presented our DEL framework where a fragment-based VAE is integrated such that evolutionary exploration is conducted in the continuous latent representation space rather than the discrete structural space. Our intensive experiments show that DEL is able to generate novel populations of molecules with improved properties, and outperforms state-of-the-art multi-objective Bayesian optimization algorithms. Applications of DEL are certainly not restricted to design of small molecules. As future work, DEL will be tested on different datasets, other design problems, and more specific applications. Other types of DGMs and search strategies will be explored to further enhance DEL. New MOBO algorithms for latent-space based optimization need to be studied further to address issues such as scalability, unknown invalid domains in latent space, and the curse of dimensionality. Github link of our PyTorch and BoTorch implementation will be available.

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

# A APPENDIX

## A.1 DEL ALGORITHM

---

**Algorithm 1:** DEL Algorithm

---

**Inputs:** $X_0$, $Y_0$: training samples and corresponding properties;
**Result:** population of designed molecules; learned DGM;

1 **for** *each evolutionary generation* **do**
2    **if** *first generation* **then**
3      $X = X_0; Y = Y_0$;
4      Train DGM using $\{X, Y\}$ ;        // use all training data to learn DGM
5      $X, Y = \text{subset}(X, Y, M)$ ;       // a subset of $M$ examples from the original training data
6    **else**
7      Train DGM using $\{X, Y\}$ ;   // use population to further train DGM
8    **end**
9   **end**
10   $Z = \text{Encoder}(X)$ ;     // obtain latent representations of samples in $X$
11   $r, \mathcal{F} = \text{nondominated\_sort}(Y)$ ;    // $r$: non-domination ranks, $\mathcal{F}$: Pareto frontiers
12   $d = \text{compute\_crowding\_distance}(Y, \mathcal{F})$;
13   $Z' = \text{EvOp}(Z, r, d)$ ;          // evolutionary operations: selection, recombination and mutation
14   $X' = \text{Decoder}(Z')$ ;                     // generate novel samples
15   $Y' = \text{get\_property}(X')$ ;    // obtain properties of the generated samples using a simulator, e.g. RDKit
16   $X, Y = \text{form\_new\_population}(X, Y, X', Y', M)$ ;     // keep the top $M$ good samples from the combination of $\{X, Y\}$ and $\{X', Y'\}$
17 **end**
18 Return $X, Y$, DGM

---

## A.2 BUG CORRECTION

In the VAE implementation by Podda et al. (2020), the incorrect use of `view` function makes the forward flow of information messed up among different samples in a batch.

Listing 1: Wrong use of view function in the original VAE model in (Podda et al., 2020).

```
_, state = self.rnn(packed, state)
# num_layers by batch by hidden_size -> batch by hidden_layer * hidden_size
state = state.view(batch_size, self.hidden_size * self.hidden_layers)
mean = self.rnn2mean(state) # mean: batch by latent_size
logvar = self.rnn2logv(state) # logv: batch by latent_size
```

Listing 2: Our correction.

```
_, state = self.rnn(packed, state)
# num_layers by batch by hidden_size -> batch by num_layers by hidden_size
state = state.transpose(1,0)
state = state.flatten(start_dim=1) # batch by hidden_layer * hidden_size
mean = self.rnn2mean(state) # mean: batch by latent_size
logvar = self.rnn2logv(state) # logv: batch by latent_size
```

## A.3 HYPERPARAMETER SETTING FOR FRAGVAE

When not specified in the main text, the following values of hyperparameters are used in experiments.

- size of the embedding layer: 128

- mask low-frequency fragments: Yes
- masking frequency: 2
- number of low frequency clusters: 5
- window for word2vec embedding: 3
- number of recurrent neurons per layer: 128
- number of recurrent layers: 2
- size of the VAE latent space: 64
- maximum length of the sampled sequence: 10
- batch size: 128
- shuffle batches: Yes
- number of epochs to train: 50
- learning rate: 0.0005
- dropout for the recurrent layers: 0.3
- annealing step size for the scheduler: 4
- annealing rate for the scheduler: 0.8
- threshold to clip the gradient norm: 5
- number of layers in MLP for properties: 2
- number of hidden units in each hidden layer of MLP: 64
- weight on property regression loss ($\alpha$): 1
- weight on KL divergence ($\beta$): {0.01, 0.1, 1}

### A.4 HYPERPARAMETER SETTING FOR DEL

When not specified in the main text, the following values of hyperparameters are used in experiments.

- number of evolutionary generations: 10
- popularization size: {20000 (default), 100000}
- number of epochs in the initial training of DGM: 50
- annealing step size for the scheduler in initial training of DGM: 4
- annealing rate for the scheduler: 0.8
- number of epochs in a subsequent training of DGM: 30
- learning rate in the initial training of DGM: 0.0005
- annealing step size for the scheduler in initial training: 2
- probability in tournament selection: 0.95
- crossover method: {linear (default), discrete}
- mutation rate: 0.01

### A.5 HYPERPARAMETER SETTING FOR MOBO

When not specified in the main text, the following values of hyperparameters are used in experiments.

- number of initial data points: 1000
- number of batches: 30
- batch size: 8
- number of MC samples: 128

### A.6 ADDITIONAL TABLES

Table 2: Performance of FragVAE in comparison with existing methods.

| Model | Data | Validity (SMILES) | Validity (Fragments) | Novelty | Diversity |
|---|---|---|---|---|---|
| ChemVAE | ZINC | 0.170 | - | 0.980 | 0.310 |
| GrammarVAE | ZINC | 0.310 | - | 1.000 | 0.108 |
| SDVAE | ZINC | 0.435 | - | - | - |
| GraphVAE | ZINC | 0.140 | - | 1.000 | 0.316 |
| CGVAE | ZINC | 1.000 | - | 1.000 | 0.998 |
| NeVAE | ZINC | 1.000 | - | 0.999 | 1.000 |
| FragVAE ($\beta$=1.00) | ZINC | 1.000 | 0.922 | 1.000 | 0.961 |
| FragVAE ($\beta$=0.10) | ZINC | 1.000 | 0.953 | 0.999 | 0.985 |
| FragVAE ($\beta$=0.01) | ZINC | 1.000 | 0.655 | 0.997 | 0.809 |
| FragVAE ($\beta$=1.00) | PCBA | 1.000 | 0.443 | 1.000 | 0.925 |
| FragVAE ($\beta$=0.10) | PCBA | 1.000 | 0.481 | 0.983 | 0.886 |
| FragVAE ($\beta$=0.01) | PCBA | 1.000 | 0.777 | 0.997 | 0.632 |

Table 3: Running time of DEL and MOBO. Since both methods involve training FragVAE, the FragVAE training time was not counted in this table. Format: Hours:Minutes:Seconds. Note: when $\beta$ is annealed to 0.4 in DEL, $\alpha$ is annealed from 1 to 4. All experiments were carried out on a Dell Precision 5820 Workstation equipped with an Intel Xeon W-2255 CPU (10C), RAM of 128GB, and a Nvidia Quadro RTX 6000 (24GB) GPU.

| Data | DEL Hyperparameter | DEL | MOBO |
|---|---|---|---|
| ZINC | $\beta = 0.1 \rightarrow 0.4$ | 3:03:49 | 25:23:01 |
| ZINC | $\beta = 0.01 \rightarrow 0.4$ | 3:08:09 | 141:49:37 |
| PCBA | $\beta = 0.1 \rightarrow 0.4$ | 3:01:32 | 31:16:18 |
| PCBA | $\beta = 0.01 \rightarrow 0.4$ | 1:50:20 | 45:31:38 |

Table 4: Non-dominated sorting of combination of Pareto fronts from DEL with and without property prediction (PP) component as latent space regularization.

| | | Pareto Front Size for Comparison | | In New Pareto Front | |
|---|---|---|---|---|---|
| Data | DEL Parameter | With PP | Without PP | With PP | Without PP |
| ZINC | $\beta = 0.1$ | 235 | 234 | 206 (87.66%) | 0 (0%) |
| PCBA | $\beta = 0.1$ | 224 | 108 | 216 (96.43%) | 0 (0%) |

Table 5: Non-dominated sorting of combination of Pareto fronts from DEL with and without DGM finetuning (FT) phrase.

| | | Pareto Front Size for Comparison | | In New Pareto Front | |
|---|---|---|---|---|---|
| Data | DEL Parameter | With FT | Without FT | With FT | Without FT |
| ZINC | $\beta = 0.1$ | 235 | 193 | 235 (100%) | 0 (0%) |
| PCBA | $\beta = 0.1$ | 224 | 214 | 199 (88.84%) | 0 (0%) |

Table 6: Non-dominated sorting of combination of Pareto fronts from DEL with linear or discrete crossover operation. Note: when $\beta$ is annealed to 0.4, $\alpha$ is annealed from 1 to 4.

| | | Pareto Front Size for Comparison | | In New Pareto Front | |
|---|---|---|---|---|---|
| Data | DEL Parameter | Linear | Discrete | Linear | Discrete |
| ZINC | $\beta = 0.01 \rightarrow 0.4$ | 200 | 181 | 196 (98.00%) | 173 (95.58%) |
| PCBA | $\beta = 0.01 \rightarrow 0.4$ | 183 | 195 | 146 (79.78%) | 184 (94.36%) |

A.7 ADDITIONAL FIGURES

Table 7: Non-dominated sorting of combination of Pareto fronts from DEL with population sizes of 20K and 100K. Note: when $\beta$ is annealed to 0.4, $\alpha$ is annealed from 1 to 4.

| | | Pareto Front Size for Comparison | | In New Pareto Front | |
|---|---|---|---|---|---|
| Data | DEL Parameter | 20K | 100K | 20K | 100K |
| ZINC | $\beta = 0.1 \rightarrow 0.4$ | 243 | 316 | 78 (32.10%) | 304 (96.20%) |
| PCBA | $\beta = 0.1 \rightarrow 0.4$ | 228 | 354 | 88 (38.60%) | 334 (94.35%) |

Table 8: Non-dominated sorting of combination Pareto fronts from DEL with different values of $\beta$. Note: when $\beta$ is annealed to 0.4, $\alpha$ is annealed from 1 to 4. Population size: 20K.

| | Pareto Front Size for Comparison | | | | In New Pareto Front | | | |
|---|---|---|---|---|---|---|---|---|
| Data | $\beta = 0.1$ | $\beta = 0.1 \rightarrow 0.4$ | $\beta = 0.01$ | $\beta = 0.01 \rightarrow 0.4$ | $\beta = 0.1$ | $\beta = 0.1 \rightarrow 0.4$ | $\beta = 0.01$ | $\beta = 0.01 \rightarrow 0.4$ |
| ZINC | 235 | 243 | 228 | 200 | 195(82.98%) | 186(76.54%) | 170(74.56%) | 143(71.50%) |
| PCBA | 224 | 228 | 189 | 183 | 169(75.45%) | 183(80.26%) | 122(64.55%) | 111(60.66%) |

Table 9: Numbers of novel molecules that satisfy properties QED $\geq 0.88$, SAS $\leq 3$, and logP $\leq 1$ in the 1st, 5th and final (10th) generations of DEL. Numbers of training molecules satisfying same conditions are also given. Note: when $\beta$ is annealed to 0.4, $\alpha$ is annealed from 1 to 4.

| Data | Hyperparameter | Train | DEL(1) | DEL(5) | DEL(F) |
|---|---|---|---|---|---|
| ZINC | $\beta = 0.1$ | 406 | 2 | 14 | 40 |
| ZINC | $\beta = 0.1 \rightarrow 0.4$ | - | 2 | 22 | 32 |
| ZINC | $\beta = 0.01$ | - | 6 | 54 | 115 |
| ZINC | $\beta = 0.01 \rightarrow 0.4$ | - | 3 | 9 | 20 |
| PCBA | $\beta = 0.1$ | 196 | 4 | 13 | 27 |
| PCBA | $\beta = 0.1 \rightarrow 0.4$ | - | 10 | 33 | 52 |
| PCBA | $\beta = 0.01$ | - | 0 | 8 | 16 |
| PCBA | $\beta = 0.01 \rightarrow 0.4$ | - | 4 | 9 | 10 |

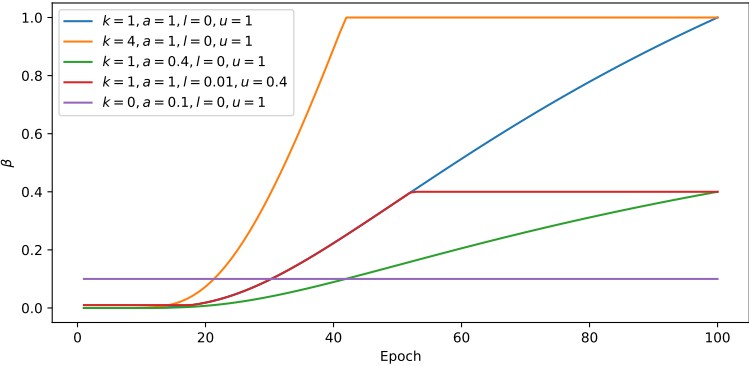

Figure 6: Different of forms of the $\beta$-function. The total number of epochs is $T = 100$.

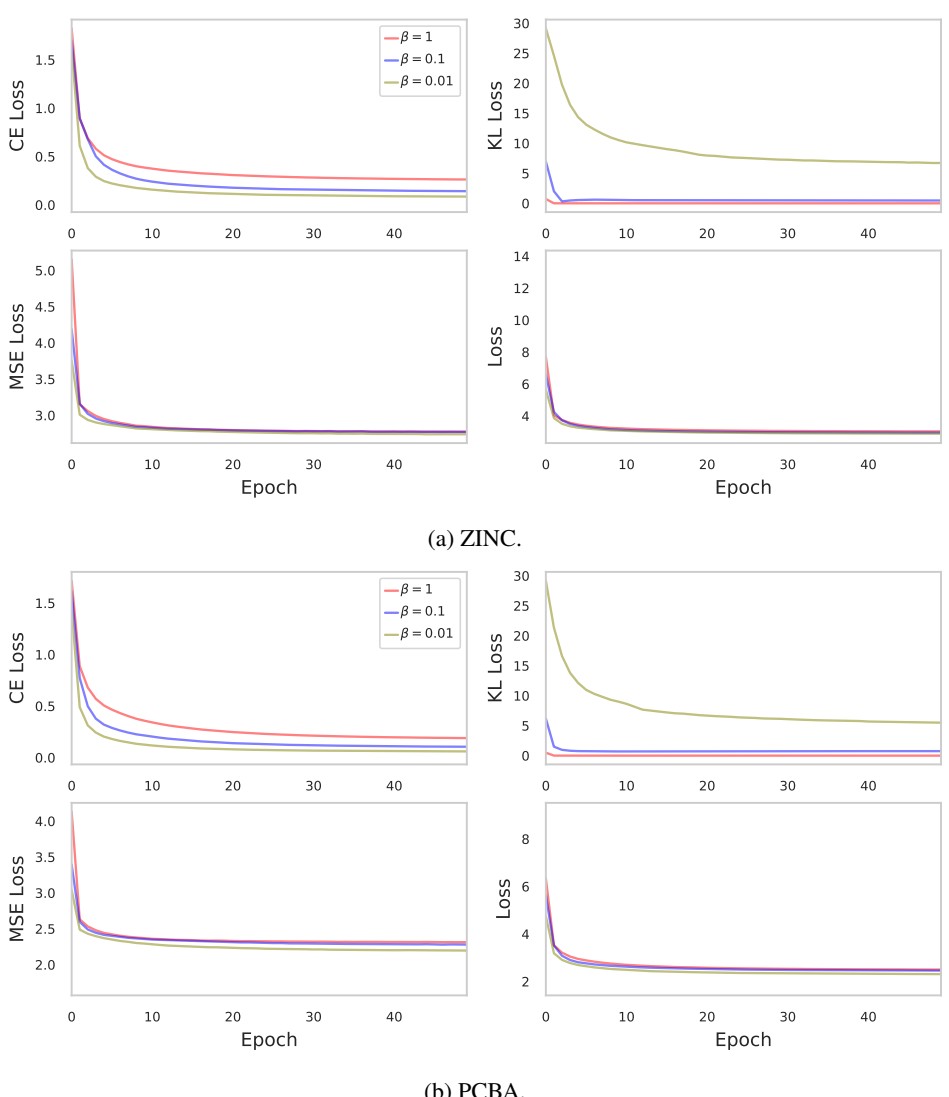

(a) ZINC.

(b) PCBA.

Figure 7: Loss of FragVAE in initial training on ZINC and PCBA respectively.

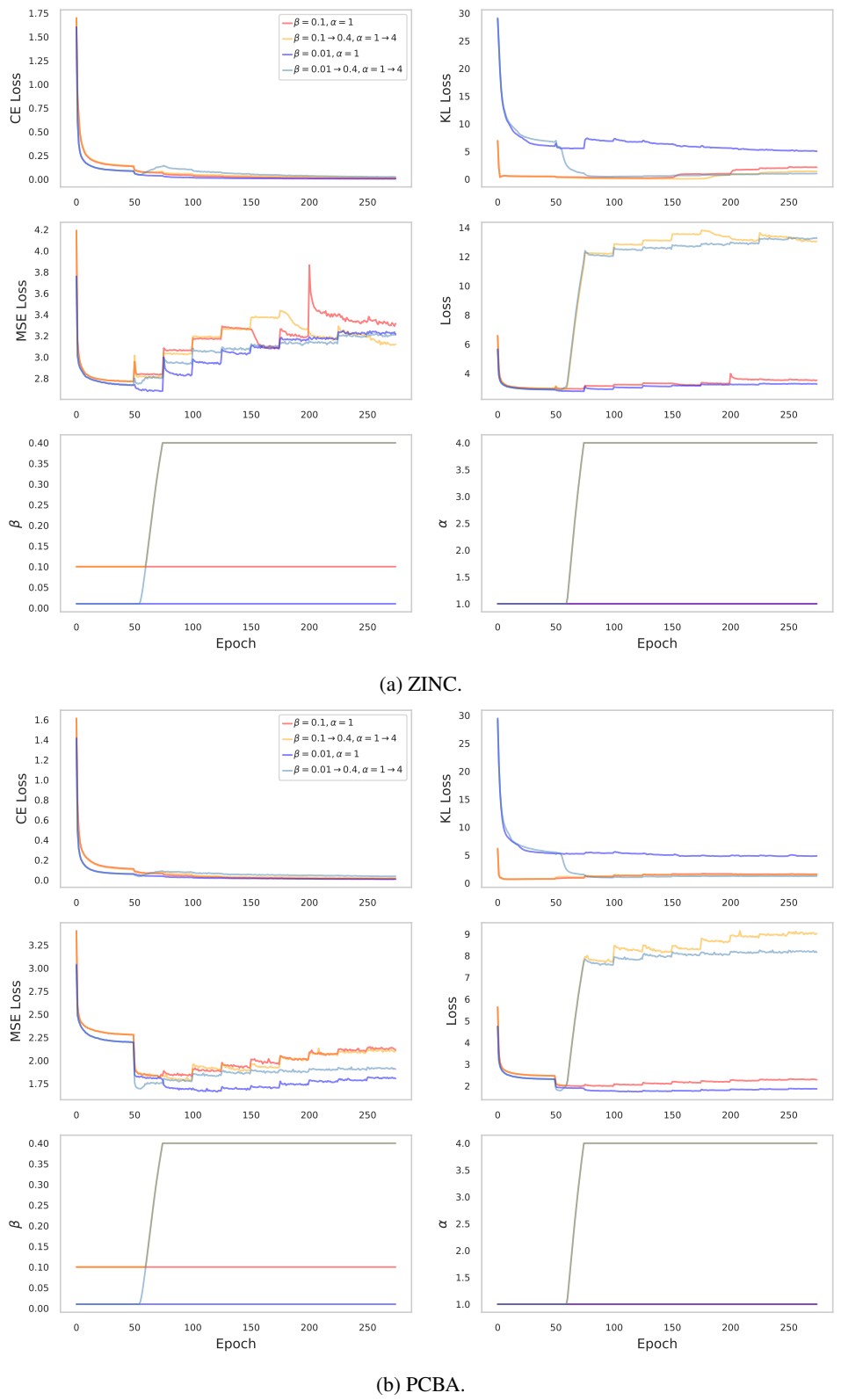

(a) ZINC.

(b) PCBA.

Figure 8: Loss of of FragVAE in DEL on ZINC and PCBA respectively.

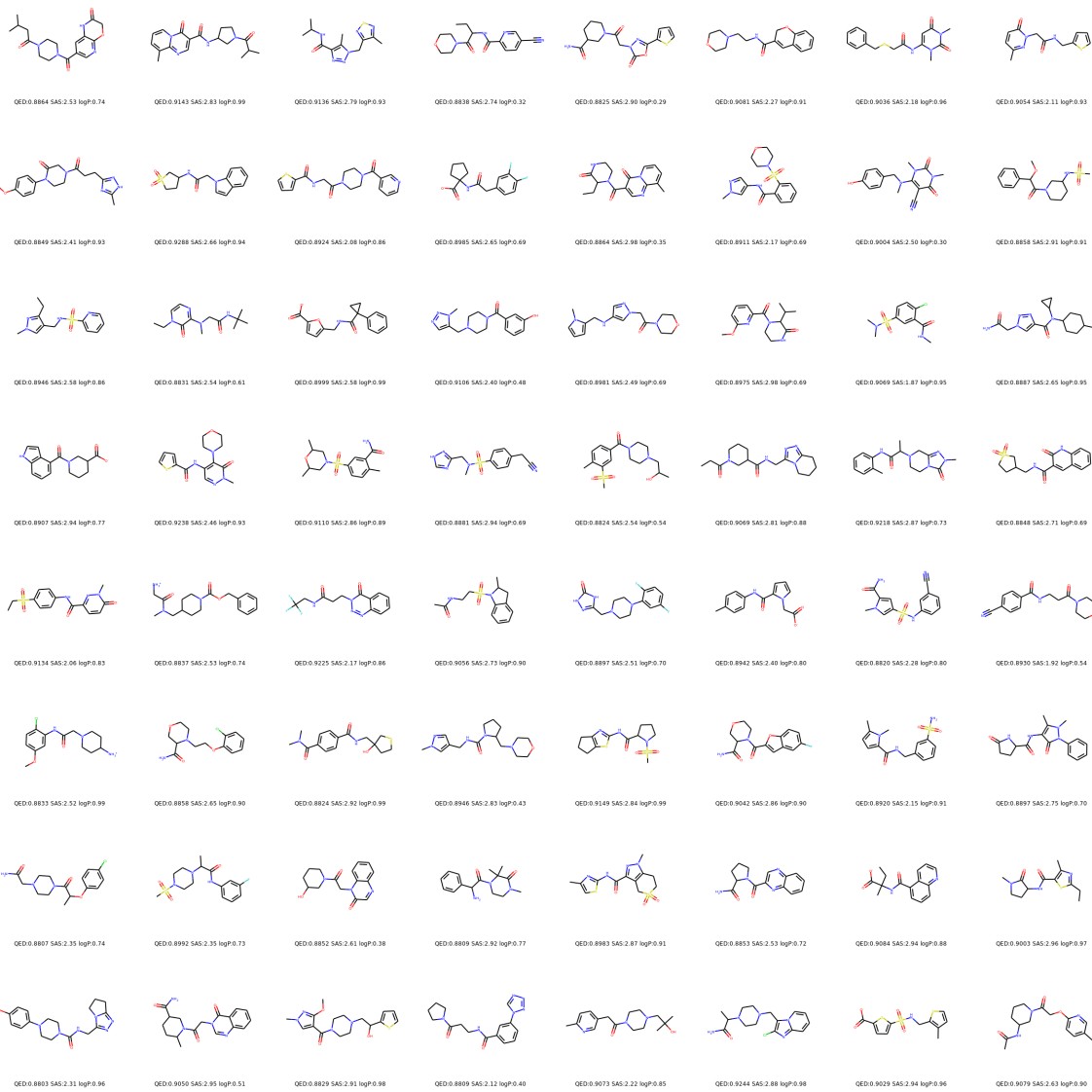

Figure 9: ZINC training molecules that satisfy properties QED≥ 0.88, SAS≤ 3, and logP≤ 1. Note: 196 molecules satisfy these conditions, but 64 are visualized.

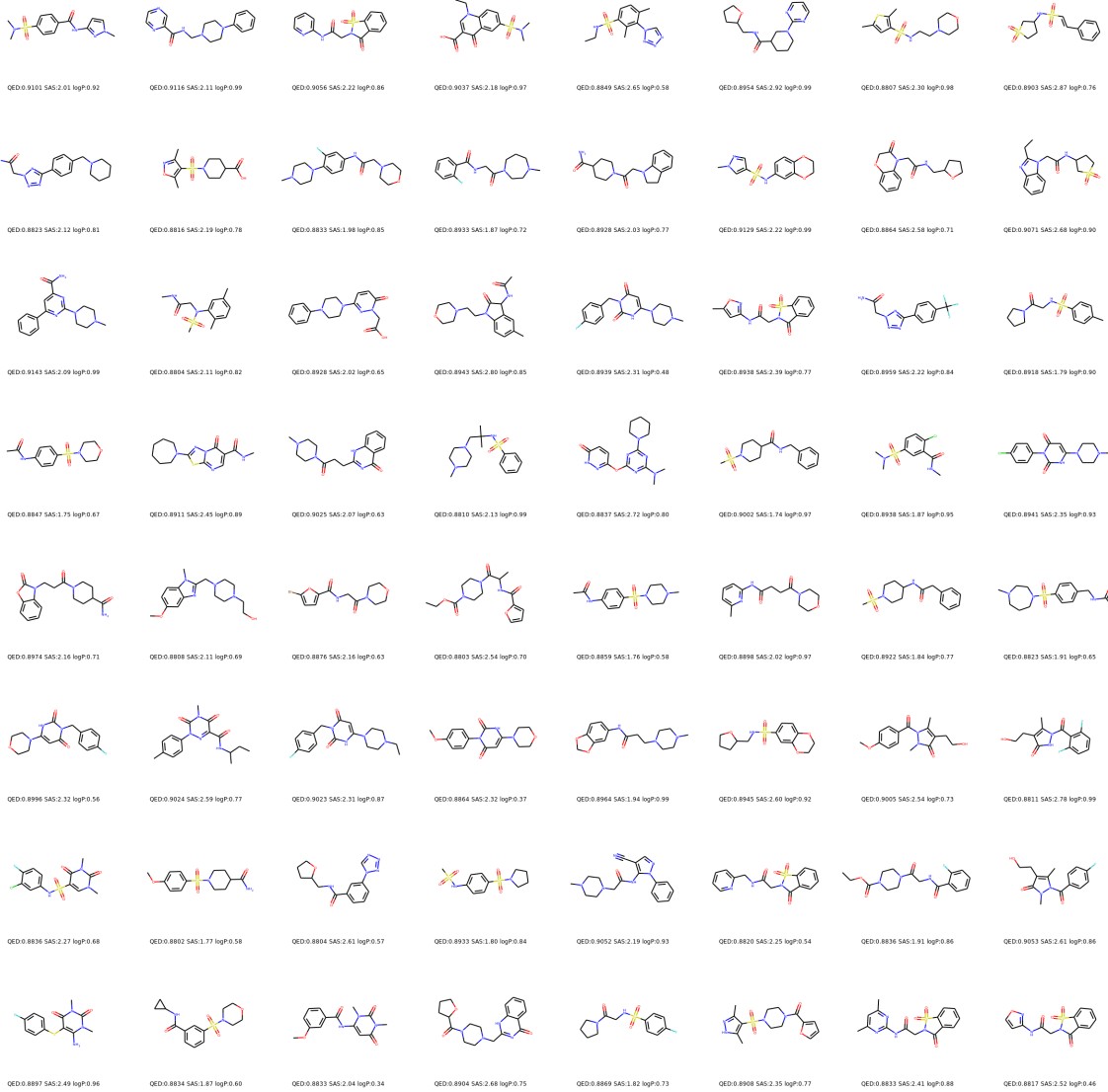

Figure 10: PCBA training molecules that satisfy properties QED $\geq 0.88$, SAS $\leq 3$, and logP $\leq 1$. Note: 406 molecules satisfy these conditions, but 64 are visualized.

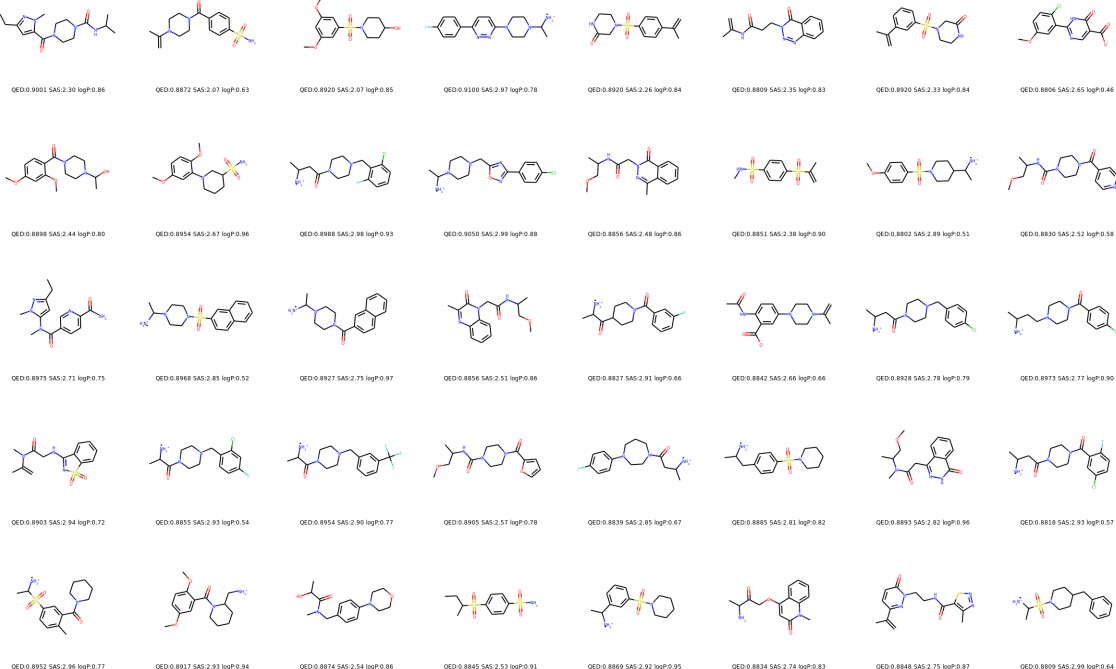

Figure 11: Novel molecules that satisfy properties QED $\geq 0.88$, SAS $\leq 3$, and logP $\leq 1$ in the final generation of DEL trained on ZINC with hyperparameter: $\beta = 0.1$, $\alpha = 1$.

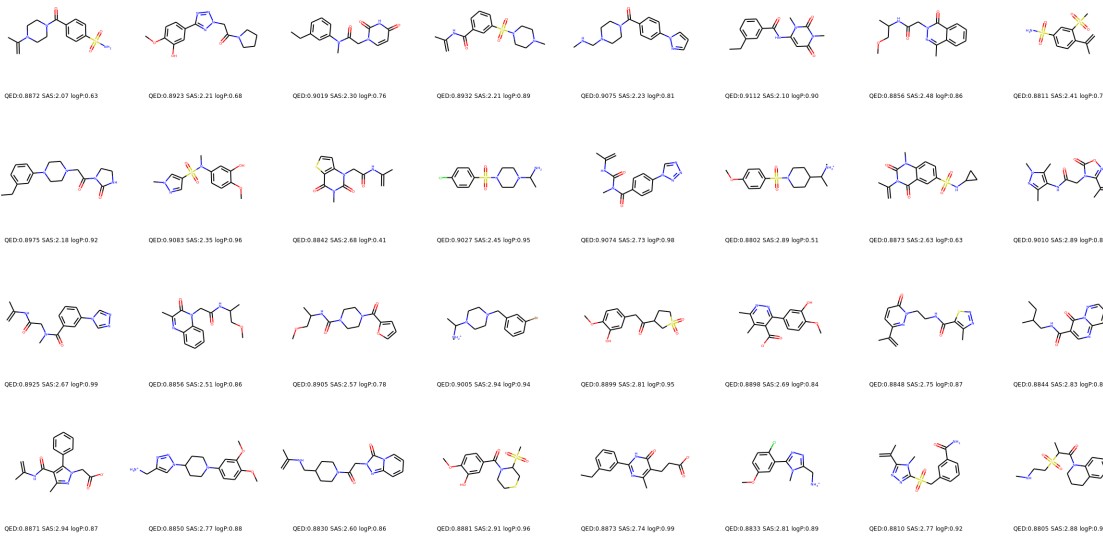

Figure 12: Novel molecules that satisfy properties QED $\geq 0.88$, SAS $\leq 3$, and logP $\leq 1$ in the final generation of DEL trained on ZINC with hyperparameter: $\beta = 0.1 \rightarrow 0.4$, $\alpha = 1 \rightarrow 4$.

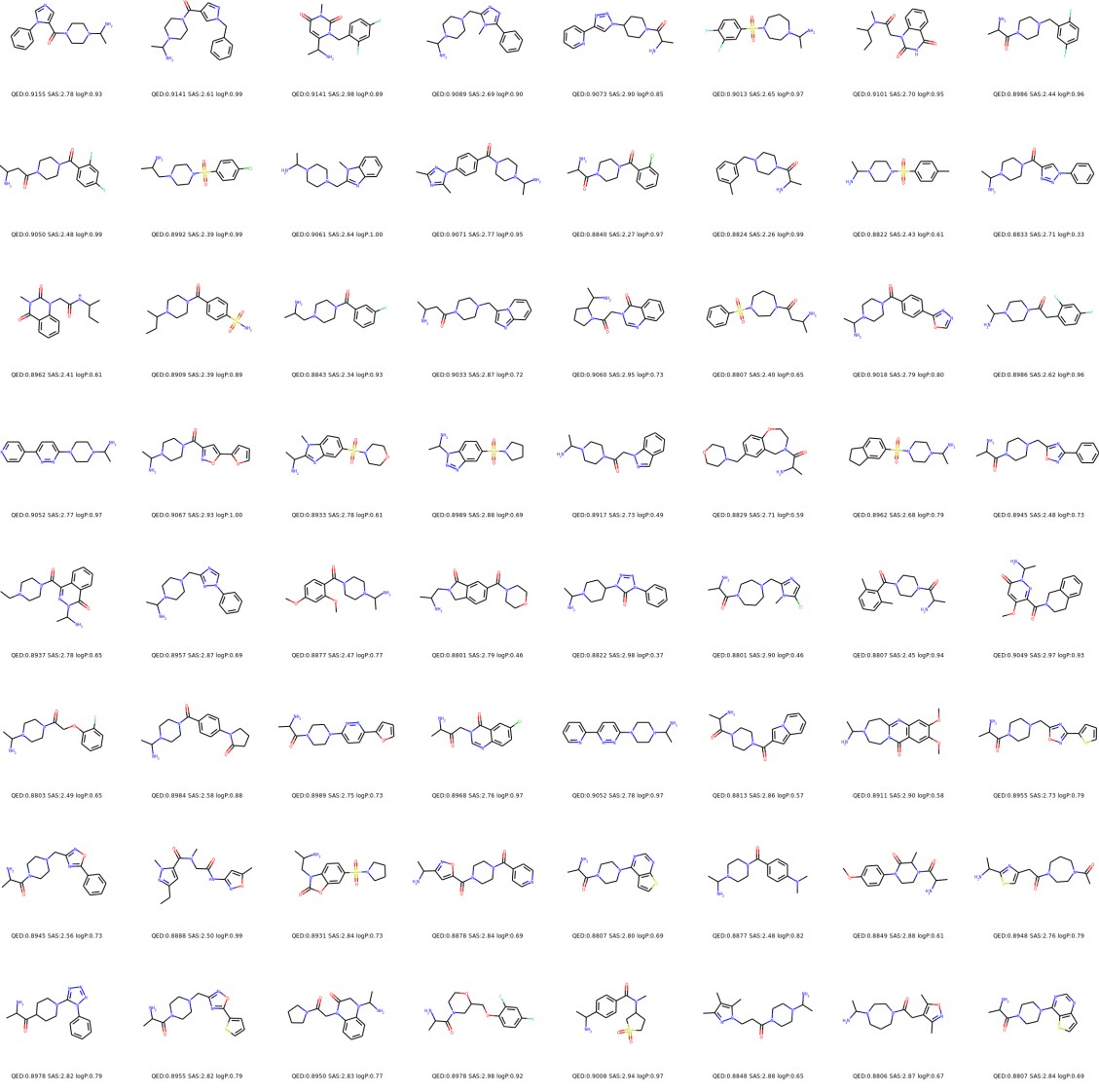

Figure 13: Novel molecules that satisfy properties QED $\geq$ 0.88, SAS $\leq$ 3, and logP $\leq$ 1 in the final generation of DEL trained on ZINC with hyperparameter: $\beta = 0.01$, $\alpha = 1$.

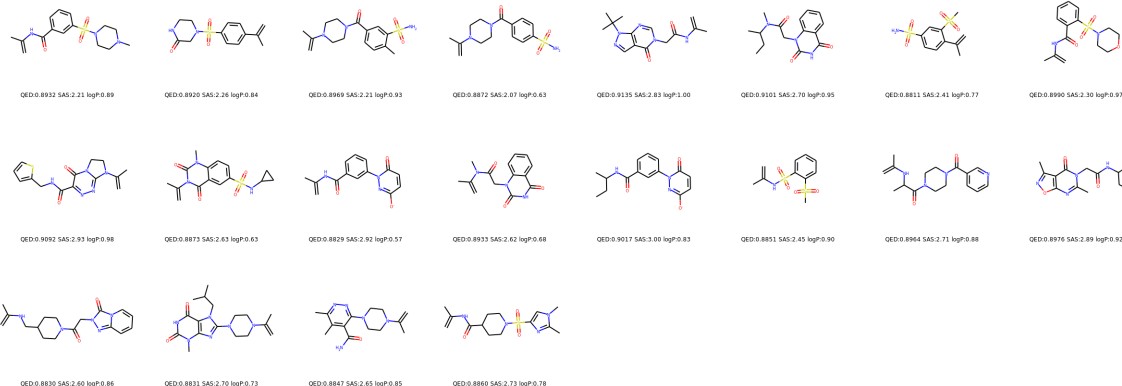

Figure 14: Novel molecules that satisfy properties QED $\geq 0.88$, SAS $\leq 3$, and logP $\leq 1$ in the final generation of DEL trained on ZINC with hyperparameter: $\beta = 0.01 \rightarrow 0.4$, $\alpha = 1 \rightarrow 4$.

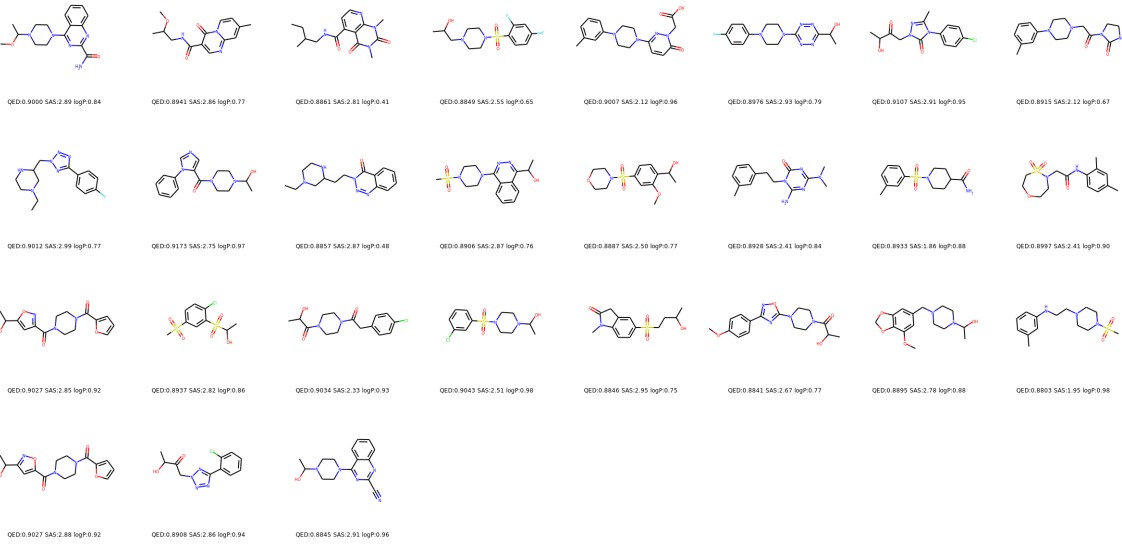

Figure 15: Novel molecules that satisfy properties QED $\geq 0.88$, SAS $\leq 3$, and logP $\leq 1$ in the final generation of DEL trained on PCBA with hyperparameter: $\beta = 0.1$, $\alpha = 1$.

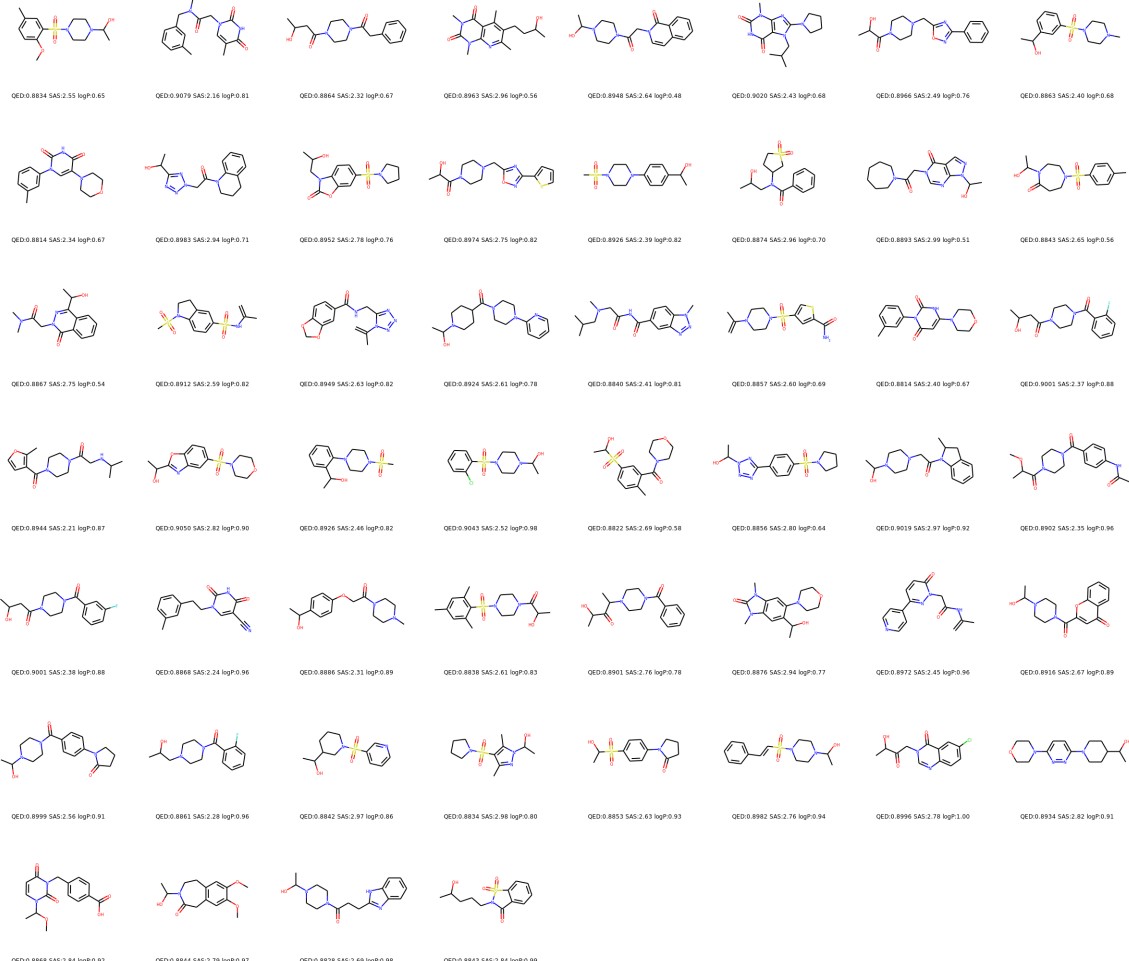

Figure 16: Novel molecules that satisfy properties QED $\geq$ 0.88, SAS $\leq$ 3, and logP $\leq$ 1 in the final generation of DEL trained on PCBA with hyperparameter: $\beta = 0.1 \rightarrow 0.4$, $\alpha = 1 \rightarrow 4$.

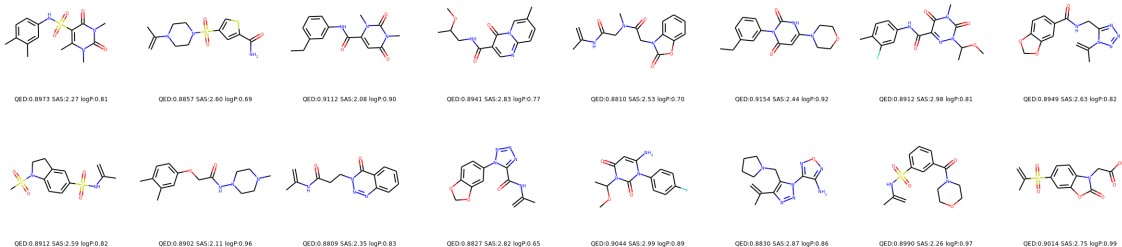

Figure 17: Novel molecules that satisfy properties QED $\geq$ 0.88, SAS $\leq$ 3, and logP $\leq$ 1 in the final generation of DEL trained on PCBA with hyperparameter: $\beta = 0.01$, $\alpha = 1$.

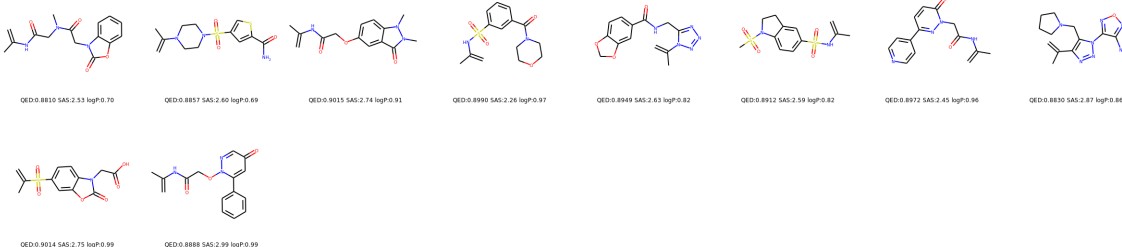

Figure 18: Novel molecules that satisfy properties QED $\geq 0.88$, SAS $\leq 3$, and logP $\leq 1$ in the final generation of DEL trained on PCBA with hyperparameter: $\beta = 0.01 \rightarrow 0.4$, $\alpha = 1 \rightarrow 4$.

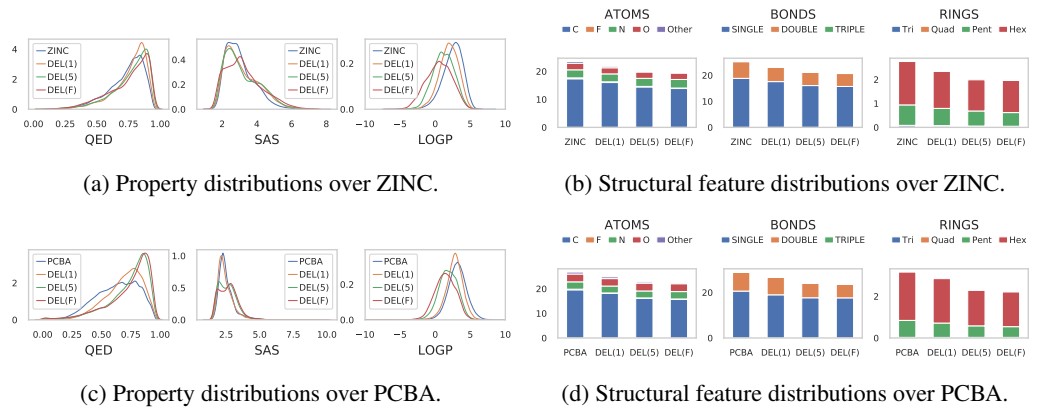

(a) Property distributions over ZINC.

(b) Structural feature distributions over ZINC.

(c) Property distributions over PCBA.

(d) Structural feature distributions over PCBA.

Figure 19: Property and structural feature distributions of population samples during DEL ($\beta = 0.1$).

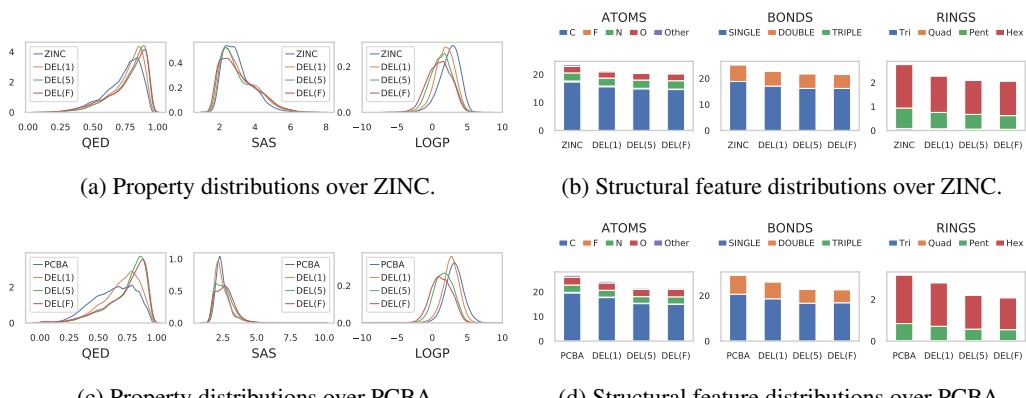

(a) Property distributions over ZINC.

(b) Structural feature distributions over ZINC.

(c) Property distributions over PCBA.

(d) Structural feature distributions over PCBA.

Figure 20: Property and structural feature distributions of population samples during DEL ($\beta = 0.1 \rightarrow 0.4$).

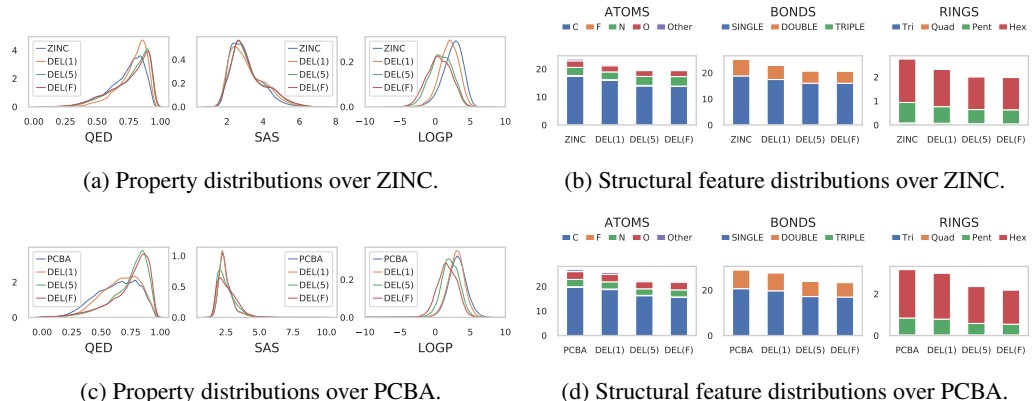

(a) Property distributions over ZINC.

(b) Structural feature distributions over ZINC.

(c) Property distributions over PCBA.

(d) Structural feature distributions over PCBA.

Figure 21: Property and structural feature distributions of population samples during DEL ($\beta = 0.01$).

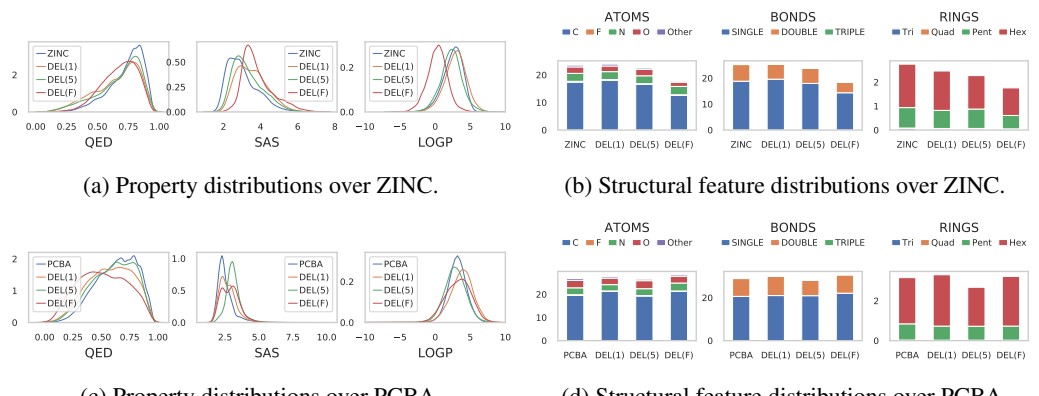

(a) Property distributions over ZINC.

(b) Structural feature distributions over ZINC.

(c) Property distributions over PCBA.

(d) Structural feature distributions over PCBA.

Figure 22: Property and structural feature distributions of randomly sampled molecules using Frag-VAE during DEL ($\beta = 0.1$).

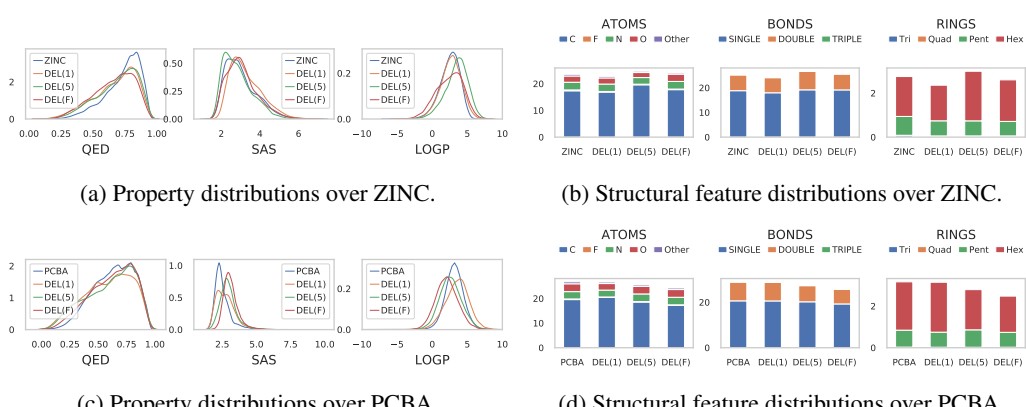

(a) Property distributions over ZINC.

(b) Structural feature distributions over ZINC.

(c) Property distributions over PCBA.

(d) Structural feature distributions over PCBA.

Figure 23: Property and structural feature distributions of randomly sampled molecules using Frag-VAE during DEL ($\beta = 0.1 \rightarrow 0.4$).

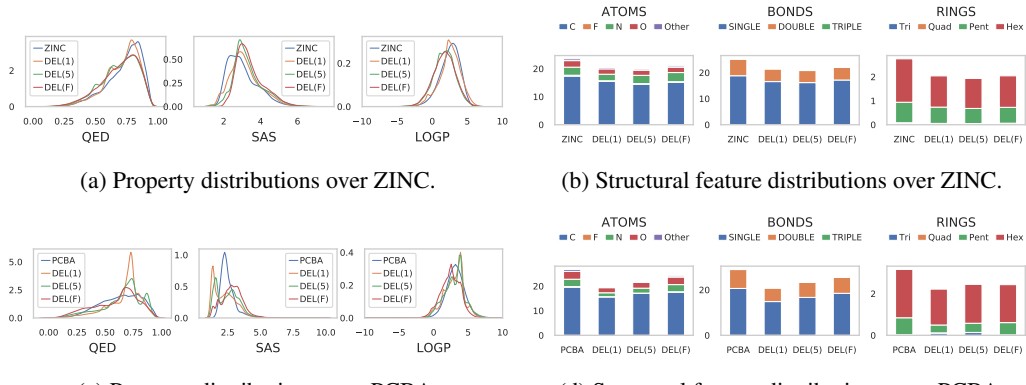

(a) Property distributions over ZINC.

(b) Structural feature distributions over ZINC.

(c) Property distributions over PCBA.

(d) Structural feature distributions over PCBA.

Figure 24: Property and structural feature distributions of randomly sampled molecules using Frag-VAE during DEL ($\beta = 0.01$).

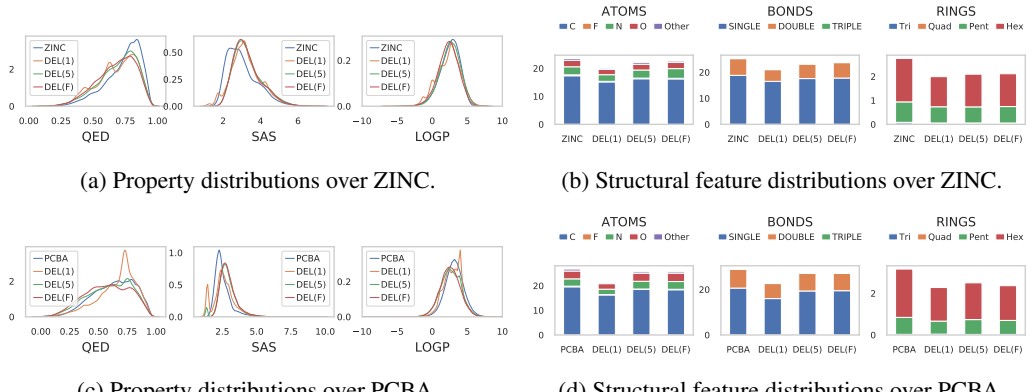

(a) Property distributions over ZINC.

(b) Structural feature distributions over ZINC.

(c) Property distributions over PCBA.

(d) Structural feature distributions over PCBA.

Figure 25: Property and structural feature distributions of randomly sampled molecules using Frag-VAE during DEL ($\beta = 0.01 \rightarrow 0.4$).

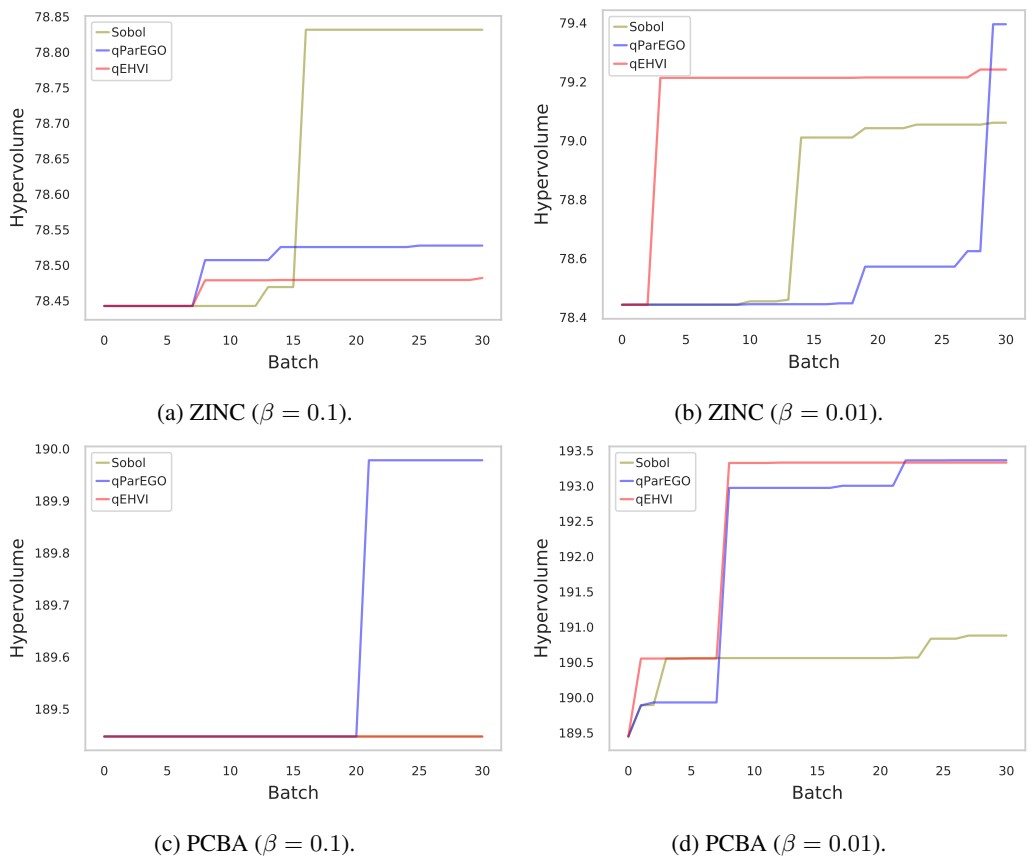

(a) ZINC ($\beta = 0.1$).

(b) ZINC ($\beta = 0.01$).

(c) PCBA ($\beta = 0.1$).

(d) PCBA ($\beta = 0.01$).

Figure 26: Hypervolume along batches when running Sobol random search, qParEGO, and qEHVI.

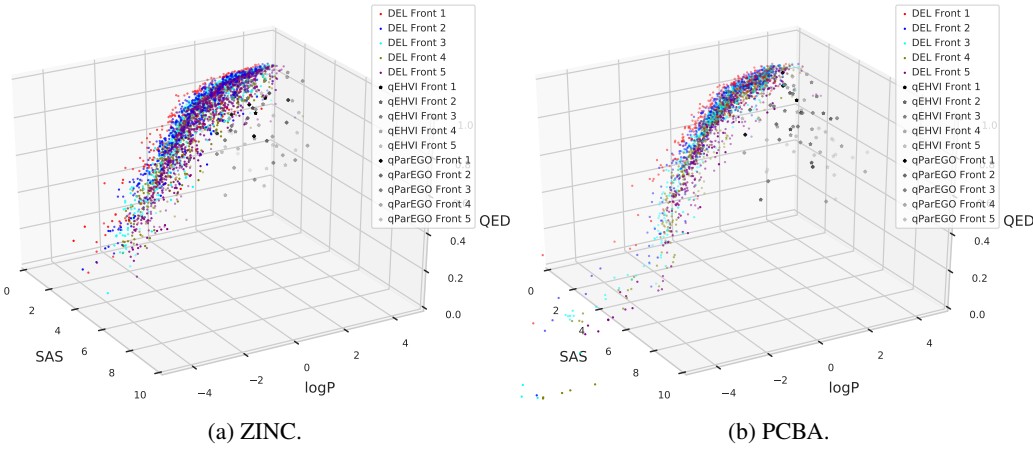

(a) ZINC.

(b) PCBA.

Figure 27: Pareto fronts of DEL and MOBO algorithms ($\beta = 0.1$). In the legend, the last batch of qParEGO or qEHVI is written as Front 1.

| Data | Property Predictor | Finetune DGM | Crossover | Population Size | beta | alpha | Evolutionary Generation | Validity (SMILES) | Validity (Fragments) | Novelty | Diversity |
|---|---|---|---|---|---|---|---|---|---|---|---|
| ZINC | Y | Y | linear | 20K | 0.10 | 1 | 1 | 1.000 | 0.873 | 0.997 | 0.988 |
| | Y | Y | linear | 20K | 0.10 | 1 | 5 | 1.000 | 0.961 | 0.999 | 0.921 |
| | Y | Y | linear | 20K | 0.10 | 1 | 10 | 1.000 | 0.554 | 1.000 | 0.805 |
| | N | Y | linear | 20K | 0.10 | 1 | 1 | 1.000 | 0.986 | 1.000 | 0.960 |
| | N | Y | linear | 20K | 0.10 | 1 | 5 | 1.000 | 0.539 | 1.000 | 0.989 |
| | N | Y | linear | 20K | 0.10 | 1 | 10 | 1.000 | 0.971 | 1.000 | 0.920 |
| | Y | N | linear | 20K | 0.10 | 1 | 1 | 1.000 | 0.936 | 0.999 | 0.977 |
| | Y | N | linear | 20K | 0.10 | 1 | 5 | 1.000 | 0.947 | 0.999 | 0.978 |
| | Y | N | linear | 20K | 0.10 | 1 | 10 | 1.000 | 0.948 | 0.999 | 0.975 |
| | Y | Y | linear | 20K | 0.1 to 0.4 | 1 to 4 | 1 | 1.000 | 0.957 | 0.999 | 0.986 |
| | Y | Y | linear | 20K | 0.1 to 0.4 | 1 to 4 | 5 | 1.000 | 0.259 | 0.999 | 0.986 |
| | Y | Y | linear | 20K | 0.1 to 0.4 | 1 to 4 | 10 | 1.000 | 0.396 | 0.996 | 0.986 |
| | Y | Y | linear | 100K | 0.1 to 0.4 | 1 to 4 | 1 | 1.000 | 0.706 | 0.996 | 0.880 |
| | Y | Y | linear | 100K | 0.1 to 0.4 | 1 to 4 | 5 | 1.000 | 0.432 | 1.000 | 0.334 |
| | Y | Y | linear | 100K | 0.1 to 0.4 | 1 to 4 | 10 | 1.000 | 0.591 | 0.999 | 0.436 |
| | Y | Y | linear | 20K | 0.01 | 1 | 1 | 1.000 | 0.819 | 0.995 | 0.956 |
| | Y | Y | linear | 20K | 0.01 | 1 | 5 | 1.000 | 0.945 | 0.998 | 0.958 |
| | Y | Y | linear | 20K | 0.01 | 1 | 10 | 1.000 | 0.973 | 0.999 | 0.963 |
| | Y | Y | linear | 20K | 0.01 to 0.4 | 1 to 4 | 1 | 1.000 | 0.917 | 0.996 | 0.911 |
| | Y | Y | linear | 20K | 0.01 to 0.4 | 1 to 4 | 5 | 1.000 | 0.940 | 0.999 | 0.978 |
| | Y | Y | linear | 20K | 0.01 to 0.4 | 1 to 4 | 10 | 1.000 | 0.919 | 1.000 | 0.976 |
| | Y | Y | discrete | 20K | 0.01 to 0.4 | 1 to 4 | 1 | 1.000 | 0.888 | 0.997 | 0.887 |
| | Y | Y | discrete | 20K | 0.01 to 0.4 | 1 to 4 | 5 | 1.000 | 0.974 | 0.995 | 0.934 |
| | Y | Y | discrete | 20K | 0.01 to 0.4 | 1 to 4 | 10 | 1.000 | 0.966 | 0.997 | 0.904 |
| PCBA | Y | Y | linear | 20K | 0.10 | 1 | 1 | 1.000 | 0.513 | 0.994 | 0.976 |
| | Y | Y | linear | 20K | 0.10 | 1 | 5 | 1.000 | 0.787 | 0.999 | 0.981 |
| | Y | Y | linear | 20K | 0.10 | 1 | 10 | 1.000 | 0.839 | 0.993 | 0.976 |
| | N | Y | linear | 20K | 0.10 | 1 | 1 | 1.000 | 0.467 | 1.000 | 0.987 |
| | N | Y | linear | 20K | 0.10 | 1 | 5 | 1.000 | 0.983 | 0.999 | 0.912 |
| | N | Y | linear | 20K | 0.10 | 1 | 10 | 1.000 | 0.907 | 1.000 | 0.818 |
| | Y | N | linear | 20K | 0.10 | 1 | 1 | 1.000 | 0.453 | 1.000 | 0.990 |
| | Y | N | linear | 20K | 0.10 | 1 | 5 | 1.000 | 0.444 | 1.000 | 0.989 |
| | Y | N | linear | 20K | 0.10 | 1 | 10 | 1.000 | 0.433 | 1.000 | 0.990 |
| | Y | Y | linear | 20K | 0.1 to 0.4 | 1 to 4 | 1 | 1.000 | 0.527 | 0.980 | 0.935 |
| | Y | Y | linear | 20K | 0.1 to 0.4 | 1 to 4 | 5 | 1.000 | 0.984 | 0.994 | 0.918 |
| | Y | Y | linear | 20K | 0.1 to 0.4 | 1 to 4 | 10 | 1.000 | 0.694 | 0.999 | 0.915 |
| | Y | Y | linear | 100K | 0.1 to 0.4 | 1 to 4 | 1 | 1.000 | 0.783 | 0.996 | 0.799 |
| | Y | Y | linear | 100K | 0.1 to 0.4 | 1 to 4 | 5 | 1.000 | 0.808 | 1.000 | 0.897 |
| | Y | Y | linear | 100K | 0.1 to 0.4 | 1 to 4 | 10 | 1.000 | 0.526 | 1.000 | 0.896 |
| | Y | Y | linear | 20K | 0.01 | 1 | 1 | 1.000 | 0.301 | 0.997 | 0.417 |
| | Y | Y | linear | 20K | 0.01 | 1 | 5 | 1.000 | 0.912 | 1.000 | 0.655 |
| | Y | Y | linear | 20K | 0.01 | 1 | 10 | 1.000 | 0.762 | 1.000 | 0.798 |
| | Y | Y | linear | 20K | 0.01 to 0.4 | 1 to 4 | 1 | 1.000 | 0.849 | 0.998 | 0.633 |
| | Y | Y | linear | 20K | 0.01 to 0.4 | 1 to 4 | 5 | 1.000 | 0.827 | 1.000 | 0.863 |
| | Y | Y | linear | 20K | 0.01 to 0.4 | 1 to 4 | 10 | 1.000 | 0.755 | 1.000 | 0.895 |
| | Y | Y | discrete | 20K | 0.01 to 0.4 | 1 to 4 | 1 | 1.000 | 0.670 | 0.996 | 0.318 |
| | Y | Y | discrete | 20K | 0.01 to 0.4 | 1 to 4 | 5 | 1.000 | 0.838 | 0.998 | 0.566 |
| | Y | Y | discrete | 20K | 0.01 to 0.4 | 1 to 4 | 10 | 1.000 | 0.869 | 0.996 | 0.568 |

Figure 28: Validity, novelty and diversity of population samples (after evolutionary operations and before merging with previous population) in different variants of DEL.

