# OpenReview forum: "Deep Evolutionary Learning for Molecular Design"
_ICLR.cc/2021/Conference — Reject_

### Official Review · AnonReviewer3 · 2020-10-25
**Limited novelty and contribution**

**Rating:** 4
**Confidence:** 3

**Review:**

The authors combine deep generative models and multi-objective evolutionary computation for computer-aided molecular design. In particular, they employ a variational autoencoder (FragVAE, as molecule modeler) and a multilayer perceptron (as property predictor). Evolutionary operations explore the latent space of the generative model to produce novel competitive molecules. The paper is interesting and tackle a very relevant problem. However, I detect the following severe limitation:

- My main concern is that I have the impression that this research is quite incremental and may not be suitable for publication in a top-tier machine learning conference. We should not forget that Evolutionary Computation techniques have been extensively applied to computer-aided molecular design and bioinformatics. In fact, there are works on the subject already published in the 90's (see, for instance, [1, 2, 3]). In the same way, there are already successful examples of automated drug-design applications using multi-objective evolutionary algorithms [4]. In this regard, I am not sure that the fact that the contributions presented in the paper (last paragraph of page 2) are sufficient. Furthermore, if I'm not mistaken, the authors did not design any specific evolutionary operator (selection, crossover and mutation) for this task and relied on conventional ones. In the same way, the authors adopted an already existing VAE model for molecular generation (Podda et al., 2020).

[1] Clark, David E., and David R. Westhead. "Evolutionary algorithms in computer-aided molecular design." Journal of Computer-Aided Molecular Design 10.4 (1996): 337-358.

[2] Parrill, Abby L. "Evolutionary and genetic methods in drug design." Drug Discovery Today 1.12 (1996): 514-521.

[3] Willett, Peter. "Genetic algorithms in molecular recognition and design." Trends in biotechnology 13.12 (1995): 516-521.

[4] Besnard, Jérémy, et al. "Automated design of ligands to polypharmacological profiles." Nature 492.7428 (2012): 215-220.

There are other aspects that, in my humble opinion, should be clarified or improved in the paper:

- Not clear if the properties of the generated samples are obtained using the MLP (as indicated in page 3 and Figure 1) or by the RDKit simulator (Landrum, 2006) (as also mentioned in page 3). Could the authors clarify this point?

- How all hyperparameters (FragVAE, MLP, evolutionary algorithm) were selected and how dependent are the results obtained to the values selected?

- Why only multi-objetive Bayesian optimization methods are included in the final comparison? Also, in relation to the experimental section, I think that in this final comparison the authors should employ statistical hypothesis testing to check the existence of statistically significant differences between the methods under comparison.

---

> ### Author Response · Authors · 2020-11-25
> **Some clarifications**
>
> Thanks for your comments. Please see below our responses.
>
> 1. The major novelty of our work is the apply evolutionary computing in the latent space of deep generative models, which is very unlike your mentioned methods that apply EC in the original space and did not combine with deep learning.
>
> 2. Our DEL framework is simple and flexible, thus other DGMs and EC algorithms can be easily embedded in it. Here we show a prototype of our idea and show that it works well.
>
>  3. As mentioned in line 15 of Algorithm 1, the properties of new molecules are obtained using RDKit.
>
> 4. We will continue to do comparative studies with baselines using statistical measures for evaluation.

---

### Official Review · AnonReviewer2 · 2020-10-28
**Needs major improvement**

**Rating:** 4
**Confidence:** 3

**Review:**

The authors propose to optimize the continuous representation of molecules in a latent space learned by a fragment-based variational autoencoder using an evolutionary algorithm. To improve the quality of the generated molecules over time, they use new generated samples as augmented data to fine-tune the generative model in each iteration. Experiments are conducted to demonstrate the effectiveness of the proposed algorithm compared with Bayesian optimization-based methods.

The paper paves a way for combining multi-objective evolutionary computation and deep generative modelling, which could be a potential benefit to other areas in machine learning. However, the paper is not ready for publication at ICLR, since it fails to discuss/empirically compare with highly related work, makes bold but incorrect statements about existing method, and lacks of clarity in presenting its own method.

**Related work**
Regarding multi-objective property optimization, \cite{li2018multi, jin2020composing} investigate the same problem and the latter one is one of the state-of-the-arts. Regarding evolutionary algorithm/genetic algorithm in the drug design space, \cite{Nigam2020Augmenting, leguy2020evomol} are highly relevant, and the authors should discuss how to position their work among these ones.

The second last sentence in the first page says, "When SMILES strings are used in VAE, the model suffers from imbalance of tokens in embedding, generation of invalid structures," which is incorrect. There are many work in the space to address this problem, e.g. grammar/syntax guided variational autoencoders~\cite{{kusner2017grammar, dai2018syntax, jin2018junction}.


**Presentation**
The lack of details and clarity in the method section makes the paper hard to understand. For example, symbol \bm{y} in equation (1) is never introduced in the context. Since the descriptions below mentioned "mean square error", I assume that \bm{y} represents the ground true property produced by the simulator here. But the authors never explain the intuition behind this. Why is using the mean square error produced by the property predictor is helpful for training the generative model? This is very confusing and it would be better to jusify this in the text.

The role of the property predictor also confuses me, because we have this simulator which can produce ground truth properties. Is regularizing the generative model the sole purpose of the property predictor? It seems that in section 2.2 the predictor is involved in the evolution step. But this contradicts with section 3.4 where there is a variant in which the property predictor is disabled.

**Evaluation**
Although the paper compared with Bayesian optimization in the experiment section, it's still insufficient to convince the audience. Comparison with Bayesian optimization seems like an ablation study, and it would be hard to understand its effectiveness without comparing to other, especially state-of-the-art approaches. I suggest the author at least compare with the following baselines:

1. a gradient-based baseline: continuous optimization in latent space using gradient of the learned property predictor.
2. \citep{jin2020composing}: one of the state-of-the-art in drug design, or other good-performing baselines using RL.

@article{kusner2017grammar,
  title={Grammar variational autoencoder},
  author={Kusner, Matt J and Paige, Brooks and Hern{\'a}ndez-Lobato, Jos{\'e} Miguel},
  journal={arXiv preprint arXiv:1703.01925},
  year={2017}
}

@article{dai2018syntax,
  title={Syntax-directed variational autoencoder for structured data},
  author={Dai, Hanjun and Tian, Yingtao and Dai, Bo and Skiena, Steven and Song, Le},
  journal={arXiv preprint arXiv:1802.08786},
  year={2018}
}

@inproceedings{jin2018junction,
  title={Junction Tree Variational Autoencoder for Molecular Graph Generation},
  author={Jin, Wengong and Barzilay, Regina and Jaakkola, Tommi},
  booktitle={International Conference on Machine Learning},
  pages={2323--2332},
  year={2018}
}


@article{li2018multi,
  title={Multi-objective de novo drug design with conditional graph generative model},
  author={Li, Yibo and Zhang, Liangren and Liu, Zhenming},
  journal={Journal of cheminformatics},
  volume={10},
  number={1},
  pages={33},
  year={2018},
  publisher={Springer}
}

@article{jin2020composing,
  title={Composing Molecules with Multiple Property Constraints},
  author={Jin, Wengong and Barzilay, Regina and Jaakkola, Tommi},
  journal={arXiv preprint arXiv:2002.03244},
  year={2020}
}

@inproceedings{
Nigam2020Augmenting,
title={Augmenting Genetic Algorithms with Deep Neural Networks for Exploring the Chemical Space},
author={AkshatKumar Nigam and Pascal Friederich and Mario Krenn and Alan Aspuru-Guzik},
booktitle={International Conference on Learning Representations},
year={2020},
url={https://openreview.net/forum?id=H1lmyRNFvr}
}

@article{leguy2020evomol,
  title={EvoMol: a flexible and interpretable evolutionary algorithm for unbiased de novo molecular generation},
  author={Leguy, Jules and Cauchy, Thomas and Glavatskikh, Marta and Duval, B{\'e}atrice and Da Mota, Benoit},
  journal={Journal of Cheminformatics},
  volume={12},
  number={1},
  pages={1--19},
  year={2020},
  publisher={BioMed Central}
}

Update after rebuttal:
Thanks for the response! Combining deep generative modeling with evolutionary algorithms is a very interesting idea in general. I hope the authors can continue to improve the paper (especially on the evaluation part) and resubmit in the future.

---

> ### Author Response · Authors · 2020-11-25
> **Thank you**
>
>
> Thank you for your review that can really help us improve our work. I was busy with lot of other commitments (teaching, reviewing, etc). But I will focus on improving this work from now on. Below are a few quick responses to your comments.
>
> 1. RDKit is used as simulator to calculate properties of new molecules. The property predictor is merely designed to regularize the generative model.
>
> 2. The work in [Nigam2020Augmenting] is seemingly similar to our work when reading the title. However their work neither take advantage of the latent representation space nor multi-objectivity, thus essentially very different from ours.
>
> 3. EvoMol proposed in [leguy2020evomol] is a rather classic EC algorithm on molecular graph. It does not combine with deep generative models.  Thus our method is substantially different from EvoMol.
>
> 4. We will have more comparisons with baselines including the above two methods.
>
> Sincerely,
>  --

---

### Official Review · AnonReviewer1 · 2020-10-28
**Careful tuning of existing methods, but limited technical novelty**

**Rating:** 4
**Confidence:** 4

**Review:**


= quality/ clarity =
needs work. see below.  too much focus on minor technical details (e.g. KL annealing for VAEs). not enough discussion of the more technically novel content, like the details of the multi-objective optimization method.

= originality / significance =
Draws on some interesting ideas from the evolutionary computation community on multi-objective optimization. However, these are not the focus of exposition. The VAE details overlap considerably with prior work.

Experiments are for standard tasks from recent literature, but there is limited comparison to recently published methods.

=Major Comments=

The Methods section is very dense. You should start with a definition of the problem. It was unclear, for example, that your optimization would be collecting multiple rounds of data and that you have a starting set of labeled and unlabeled data. You should definitely put an algorithm box in the main paper body and put many of the details that have been introduced in prior work (e.g. KL annealing) in an appendix.

The Methods section does not clearly delineate what is prior work and what is a novel contribution. Is there anything in Sec 2.1 novel, for example? Overall, is my assessment correct that your work combines an existing encoder-decoder model (FragVAE) with some methods from the evolution literature for multi-objective optimization? Is the primary novelty to adapt these evolution methods to work in latent space?

The difference between FragVAE and Podda et al., 2020 is minor, mostly consisting of the addition of a supervised regression network and some re-balancing of the terms in the loss, KL annealing, etc. Sec 3.1 of the results analyzes the impact of tuning these details. While this is important for a practitioner's model development, they will not be of sufficient general interest to readers. Sec 3.2 is also mostly focused again on understanding the impact of tuning parameter choice and does not compare to alternative methods from the literature.

Sec 3.3: The choice of baselines is very focused on comparing against other methods for multi-objective optimization. Why not also compare against other optimization methods for recent ML/chemistry papers that perform single-objective optimization? Many of these are referred to in the introduction.

Overall, the methods in section 2.2 for multi-objective optimization are interesting and introducing them to the modern ML community is a valid contribution. However, the paper's exposition and experiments are too focused on details of the FragVAE model and do not analyze the details of the multi-objective optimization approach enough.

---

> ### Author Response · Authors · 2020-11-25
> **Thanks for your comments**
>
> In addition to our feedback to Reviewer 4, a few specific clarifications here:
>
> 1. We do not tend to sell FragVAE. Basically most DGMs for molecular design can be instead embedded in our DEL system.
>
> 2. Our major novelty is the creation of the DEL system.
>
> 3. Within the next few days, we will continue to improve the presentation of this paper, conduct more comparisons with baselines and analysis.
>
> Thanks!

---

### Official Review · AnonReviewer4 · 2020-10-28
**Official Blind Review #4**

**Rating:** 4
**Confidence:** 4

**Review:**

Summary: The paper proposes to tackle multi-objective optimization of molecular properties by combining a genetic algorithm with a fragment-based generative model of SMILES strings. Using a generative model allows to perform evolution in the latent space, as opposed to molecule space. Experiments show the model can produce a rich Pareto front of samples, outperforming bayesian optimization ran in the latent space of the same generative model.

Recommendation: Overall, I am voting to reject. While combining evolutionary algorithms with deep molecular generative models is a very interesting research direction, the current draft uses a very limited set of metrics and baselines, making it unclear if the proposed method is comparable to the state-of-the-art. Technical novelty of the paper is rather limited (combining several previously published ideas), although the particular combination does seem novel. The approach looks promising, but with the provided set of experiments it's hard to compare it to existing methods.

Main pros:
1. The paper is easy to understand and explores a timely and important topic.
2. Using concepts from multi-objective optimization (non-domination ranks and crowding distance) sounds like a good idea, given that some established molecular optimization works simply convert the multi-objective problem into a single-objective one by weighing the different terms, and using Pareto frontiers as done in this work could uncover a much richer structure in solution space.
3. Evolution in the latent space is an interesting approach, which bears some similarities to using particle swarm optimization as in [1]. It would be interesting to see a comparison (quantitative and qualitative) of these two methods.

Main cons:
1. Most of my concerns are about weak metrics and baselines:
- For benchmarking the generative model in isolation, the authors use validity, novelty, diversity, and simple molecular distribution statistics (atom, ring, and bond types, together with distributions of simple properties). This is a good sanity check, but a rather weak metric to compare to established models. Perfect validity, novelty and diversity can be obtained with an untrained graph-based model (e.g. [3]), while the other metrics do not capture more fine grained distribution statistics. It would be useful to also compute more complex metrics, such as those in [4] (KL divergence on a larger set of chemical properties, and Frechet ChemNet Distance), and additionally benchmark reconstruction success. Of course, a weak result on any distribution matching metric does not mean the model will not perform well for optimization; nevertheless, having the metrics for FragVAE would provide more insight into the model.
- For benchmarking optimization, the authors jointly optimize three simple molecular properties. It is a bit unclear if that single task is representative of the broader class of problems encountered in generative chemistry. One could additionally optimize for the 20 tasks proposed in [4], and run quality filters on optimized samples to see if the optimizer exploited the scoring function (see [4] for the filters, and [5] for more discussion on the topic).
2. The generative model itself is the one of [2], with the addition of a property predictor (to regularize the latent space), and a warm-up scheme for the KL-divergence part of the loss. Both modifications are rather standard, which raises expectations with respect to the experimental evaluation of the method.
3. Some details of the comparison with BO are unclear:
- Depending on the comparison, molecules from either 5 or 6 BO batches are selected; how was the number of batches chosen? Given that the evolutionary process by design is likely to retain all Pareto-optimal samples seen during optimization, wouldn't it be more fair to take all samples produced by BO, instead of a fixed number of batches? The paper mentions that the BO batch size is 8 - is it the case that evolution produces 20k samples, which are then compared to 40-48 samples from BO?
- How many steps does BO do, and what is the dimension of the latent space? I am wondering what is contributing to BO being so slow (as seen in Table 3).

Other comments:
- In the loss for the property predictor, are the losses for the different kinds of properties normalized in some way?
- The paper says "One of the theoretical innovations of our approach is that it demonstrates that EC methods are extendable to corresponding deep versions." - this contribution is not theoretical; moreover, I wouldn't call the evolutionary algorithm deep (the generative model is)
- "Data evolution tends to be more efficient than direct evolution of model structures and parameters." - can the authors expand on what that means? Is that a result shown in the literature?
- In Section 2.3, the sampling of r_i could be just from Uniform(-d, 1+d), instead of linearly transforming a sample from Uniform(0, 1). Moreover, I am curious about the fact that r_i can be outside of (0, 1), which means the child may land outside of the segment connecting the parents. Did the authors observe this to work better in practice that sampling r_i from (0, 1)? One danger there is that the samples may get drawn towards the origin, was this observed empirically?
- "To clarify, the perfect validity reported in Podda et al. (2020) is actually calculated as the ratio of valid generated SMILES strings after discarding invalid fragment sequences" - can the authors expand on this? I assumed predicting the next fragment is a classification problem (with a very large number of classes), if so, how can invalid fragments be produced?
- Since using a larger population size performs better (Table 7), is there any reason this is not the default setting for the method described in the paper? Would the Pareto front keep improving with even larger population sizes?
- I cannot parse lines 2-9 of the pseudocode in Appendix A.1. Moreover, why does compute_crowding_distance take the Pareto fronts as one of the arguments?
- The bug shown in Appendix A.2 looks very serious; it seems latent codes of different samples in a batch get arbitrarily mixed up. Is it fair to say that the original implementation of [2] was mostly ignoring the latent code?
- I am confused by Table 4: as far as I understand, two groups of samples are being lumped together, and the Pareto front is recalculated. How is it possible that there are no samples from the "Without PP" setting in the combined Pareto front, yet the combined front doesn't include some samples that were Pareto-optimal in the "With PP" setting? If all samples from "Without PP" were dominated by Pareto-optimal samples from "With PP", then I would expect the combined front to be the same as for the "With PP" samples.
- What is causing the sudden jumps in the loss plots in Figure 8?

Small remarks, typos, and grammar issues (did not influence my rating recommendation):
- Page 1: "adopt" -> "adopts"
- Page 1: the paper distinguishes two classes of approaches: one based on SMILES, and one based on Graph Convolutional Neural Networks. It would be better to drop the word "Convolutional", as it does not really apply to some methods based on graph message passing (e.g. [3, 6])
- Page 1: "that, molecules," -> "that molecules,"
- Page 2: "bounds"
- Page 2: "optimizations" -> "optimization" (twice)
- Page 2: "for neuroevolution that leads to evolution" -> "for evolution"
- The EDA acronym is introduced, and never used - I would just skip introducing the acronym then
- The paper uses terms "evolutionary computation (EC)" and "evolutionary strategy (ES)", how do those differ? If they mean the same thing, I'd suggest sticking to one for clarity
- Page 2: "structural transformation" -> "structural transformations"
- Page 2: missing space in "operations.This"
- Pages 2 & 4: "for examples" -> "for example"
- Page 6: "The novelty is defined as the ratio of number of generated novel valid molecules that do not exist in the training data (...)" - I would drop the word "novel", since novelty is being defined by this very sentence
- Page 6: "proximity (...) with" -> "proximity (...) to"
- Page 13: "phrase"

References:
- [1] Efficient multi-objective molecular optimization in a continuous latent space
- [2] A Deep Generative Model for Fragment-Based Molecule Generation
- [3] Constrained Graph Variational Autoencoders for Molecule Design
- [4] GuacaMol: Benchmarking Models for de Novo Molecular Design
- [5] On Failure Modes of Molecule Generators and Optimizers
- [6] Junction Tree Variational Autoencoder for Molecular Graph Generation

----------------------------------------------------------------------------------------------------

Comments after rebuttal:

I would like to thank the authors for their response. I am keeping my score, but I encourage the authors to resubmit after improving the things that were discussed (most importantly, using better metrics and comparing to more established baselines); I think this will make the paper much stronger.

---

> ### Author Response · Authors · 2020-11-25
> **Thanks for your valuable comments**
>
> Your detailed comments are highly appreciated. In the past two weeks, I have to finish scheduled teaching and reviewing multiple papers for ICLR and AAAI. But I will focus on improving this work from now on. Below are some critical clarifications which be informative to the reviewers before the author response period is ended.
>
> 1. I will undertake more comparative studies particularly with benchmarks from GuacaMol and MOSES.
>
> 2. Our DEL framework is not restricted to the currently used DGM. Most DGMs for molecular design can be instead embedded into this framework. We will test the performance of other DGMs in DEL.
>
> 3. Our original paper is unclear. The concept "batch" used in MOBO is different from the concept "minibatch" in deep learning. "Batch" in MOBO refers to a group of data points / solutions in one iteration. As shown in Figure 26 in appendix, the two MOBO algorithms converge within 30 iterations. Putting the solutions in the last few iterations renders us 40-48 valid samples from MOBO. Compared with our method, scalability is one of the major issues. The efficiency of the two MOBO algorithms are highly restricted by the batch size (i.e. number of solutions in one iteration) and the number of MC samples in acquisition functions.
>
> 4.  In the loss for the property predictor, the losses for the different kinds of properties were not normalized. We will test the impact of property normalization.
>
> 5. "Data evolution tends to be more efficient than direct evolution of model structures and parameters." Due to high time and space complexities, it is unlikely that we can evolve many different deep generative models with limited resources. However, data evolution is highly feasible.
>
> 6. In Fragment-based molecular design, not all fragments can be assembled together due to constraints on bonds. Thus, not all generated sequences of fragments are valid.
>
> 7. Pareto fronts is needed to compute crowding distances, because the crowding distances are computed front-wise.
>
> 8. Yes, the bug in Podda et al. (2020)'s implementation is severe and crucial. We fixed it and make it really work.
>
> 9.  The use of larger population size takes longer time. It is indeed a trade-off between efficiency and performance.
>
> 10. Good question for Table 4. When we combine the Pareto fronts from two different cases and do non-dominated sorting, it is possible to have more than two new fronts.
>
> 11. Evolutionary strategy (ES) is a special family of EC algorithms.
>
> 12. The work in [Efficient multi-objective molecular optimization in a continuous latent space] is really interesting. The major difference between ours the their work is that we fine-tune the DGM using new populations of samples with better properties and show that this strategy does improve the performance in comparison with EC with static DGM.
>
> Thank you again for your comments on our work.

---

### Decision · Program_Chairs · 2021-01-07
**Final Decision**

**Decision:**

Reject

**Comment:**

This work combines deep generative models (variational autoencoders, FragVAE) and multi-objective evolutionary computation for molecular design. They use a multilayer perceptron as a predictor for properties. Evolutionary operations are used to explore the latent space of the generative model to produce novel competitive molecules. Experiments are executed to show the effectiveness of the proposed method with respect to Bayesian optimization-based methods.

Strengths:

1 - Combines multi-objective evolutionary computation and deep generative modeling, which is a promising approach to tackle multi-objective optimization in structured spaces.

Weaknesses:

All the reviewers agree that the paper is not yet ready for publication. They point out the following areas to improve:

1 - The lack of details and clarity in the method.

2 - The experimental section needs to be improved. The evaluation metrics and baselines are weak.

3 - Describe better and more clearly the novelty of the proposed approach with respect to previous work in the area.